# Predicting the protein interaction landscape of a free-living bacterium with pooled-AlphaFold3

Horia Todor [1✉], Lili M Kim [1,6], Jürgen Jänes [2,6], Hannah N Burkhart[1], Seth A Darst [3], Pedro Beltrao [2] & Carol A Gross[1,4,5]

## Abstract

**Accurate prediction of protein complex structures by AlphaFold3 and similar programs has been used to predict the presence of protein–protein interactions (PPIs), but this technique has never been applied to an entire genome due to onerous computational requirements and questionable utility. Here we present pooled-PPI prediction, a technique that dramatically improves the accuracy of genome-scale screens compared to a paired approach while simultaneously reducing inference time (~twofold) and the number of jobs (~100-fold). We use this technique to predict the structure of all 113,050 pairwise PPIs in *Mycoplasma genitalium* using only 2027 AlphaFold3 jobs. This unbiased and comprehensive dataset was highly predictive of known interactions, revealed a previously unappreciated but widespread size bias in AlphaFold interface scores, correctly identified protein–protein interfaces in macromolecular complexes, and uncovered new biology in *M. genitalium*. This work establishes pooled-PPI prediction as a highly scalable method for uncovering protein–protein interactions and a powerful addition to the functional genomics toolkit.**

**Keywords** AlphaFold3; Mycoplasma; Protein-Protein Interactions; Structural Biology
**Subject Categories** Biotechnology & Synthetic Biology; Computational Biology

## Introduction

Deciphering the function of novel proteins based on their primary sequence is a longstanding goal in biology. The release of AlphaFold2 in 2021 represented a significant step towards this goal, predicting protein structures with unprecedented accuracy (Jumper et al, 2021). However, discerning the function of a novel protein, even knowing its structure, remains a significant challenge. Both AlphaFold2 and its successor AlphaFold3 are capable of predicting the structures of protein complexes and assigning a score to predicted protein–protein interfaces in such complexes (Evans et al, 2021; Abramson et al, 2024). Associating an unannotated protein with a well-understood partner is an important and informative step to deciphering its function. Although both AlphaFold2 and AlphaFold3 were designed to predict *how* proteins interact, both have been successfully co-opted to predict *whether* two proteins interact, and an increasingly large body of evidence points to the utility of genome-scale protein–protein interaction (PPI) prediction profiling for dissecting biological processes and determining gene function (Schmid and Walter, 2025; Yu et al, 2023; Yirmiya et al, 2025; Burke et al, 2023; Schweke et al, 2024). However, the use of genome-scale PPI prediction is limited by its onerous computational requirements and the potential for false-positive hits. While there has been a significant effort to make protein structure prediction programs accessible to non-expert users (Elfmann and Stülke, 2025) (such as the web interfaces for AlphaFold3 and CHAI1 and the ColabFold notebook for AlphaFold2 and AlphaFold-Multimer), the limited nature of these interfaces makes them unsuitable for genome-scale predictions.

Here, we describe and characterize pooled-PPI prediction, a broadly applicable strategy for facilitating genome-scale screens. Relying on the fact that PPIs are rare (e.g., ~0.05% of possible protein pairs are thought to interact in humans) (Venkatesan et al, 2009), pooled-PPI prediction assumes that the predicted structure (and therefore interface score) of a true complex is unlikely to be affected by the addition of other, non-interacting proteins in the same job. Including multiple unrelated proteins (a pool) in a single job allows numerous potential pairwise interactions (*n* choose 2) to be screened simultaneously. Although mutually exclusive interactions sharing the same interface (e.g., B and B' both bind the same pocket of A) could be missed if folded together, the probability of this is low in a genome-wide approach. Pooled-PPI prediction decreases overall runtime (~twofold, depending on hardware), the number of individual jobs (up to 300-fold, depending on pool size), and increases the accuracy of PPI predictions through general competition and spatial constraints, as previously demonstrated in small scale peptide-target screens (Chang and Perez, 2023; Vosbein et al, 2024; Mondal et al, 2024).

[1]Department of Microbiology and Immunology, University of California, San Francisco, San Francisco, CA 94158, USA. [2]Institute of Molecular Systems Biology ETH Zürich, Zürich, Switzerland. [3]Laboratory of Molecular Biophysics, The Rockefeller University, New York, NY, USA. [4]Department of Cell and Tissue Biology, University of California, San Francisco, San Francisco, CA 94158, USA. [5]California Institute of Quantitative Biology, University of California, San Francisco, San Francisco 94158 CA, USA. [6]These authors contributed equally: Lili M Kim, Jürgen Jänes. ✉E-mail: horia.todor@gmail.com

We demonstrate the utility of this method by performing a comprehensive genome-wide (all by all) predicted PPI screen for the free-living bacterium *Mycoplasma genitalium* (113,050 unique protein pairs) using only the free AlphaFold3 web interface. Currently, AlphaFold3 is singularly well-suited for this approach because its computational efficiency enables high-throughput folding of large jobs (e.g., 5,000aa pools) on a single 80gb GPU, though this may change rapidly (e.g., (Litfin et al, 2025)). We make several findings. First and most importantly, we found that pooled-AlphaFold3 predictions recapitulate experimentally validated PPIs in *M. genitalium* while drastically reducing the number of false positives compared to pairwise predictions. Second, the scale and unbiased nature of our screen revealed a widespread size bias in AlphaFold3 interaction scores, which is not due to our pooled approach. Correcting for the size bias not only drastically improved the identification of biologically relevant interactions but also revealed significant variability in AlphaFold3 interface scores in both paired and pooled contexts. Third, we found that despite assessing interactions in a pairwise manner, we could reconstruct macromolecular machines such as the ribosome and RNA polymerase. Finally, we discovered novel PPIs in *M. genitalium*, including many with orthogonal experimental support, that suggest new biology. Together, these results establish pooled-PPI prediction as a powerful and broadly applicable method for facilitating genome-scale PPI screens.

# Results

## Pooled-PPI prediction improves runtime and decreases the total number of runs

Pooled-PPI prediction leverages the known rarity of PPIs (Venkatesan et al, 2009) to evaluate multiple potential PPIs in a single job by co-folding several proteins rather than only two (Fig. 1A). Since the number of potential pairwise PPIs predicted in a single job scales quadratically (as *n* choose 2) with the number of proteins (Fig. 1B), this strategy can drastically reduce the number of jobs required to predict a given set of PPIs. The number of proteins that can be included in a job depends on the protein size and the prediction algorithm. The web interface for AlphaFold3 accepts inputs up to 5000 tokens (for PPI prediction, 5000 aa) (Abramson et al, 2024). However, larger jobs require more computational time (Abramson et al, 2024; Google DeepMind, 2024) (Fig. 1C), and this scaling appears to be approximately quadratic for AlphaFold3. To determine the theoretical benefit of pooled-PPI, we considered hypothetical jobs with proteins of 200–500 aa—a range that encompasses both the median bacterial protein (~300 aa) and the median human protein (~480 aa)—and estimated the runtime of a paired and pooled approach. We found that pooling decreases total inference time by 1.4- to twofold, depending on the underlying hardware (Fig. 1D). Intuitively, this performance benefit arises from the fact that in a comprehensive pooled approach, each individual protein is folded less often than in a comparable paired approach. Importantly, the total number of jobs required decreases significantly (45- to 300-fold), allowing many more PPIs to be assayed using the free allotment of jobs provided by online resources such as the AlphaFold3 web interface (currently 30 jobs per day per user). The theoretical benefits of pooling are even more

significant for higher-order interactions—a single job with 10 proteins assesses 120 tripartite interactions while one with 25 assesses 2300 (Fig. EV1), resulting in a 16- to 25-fold improvement in overall runtime for querying such interactions.

## Prediction of all pairwise PPIs in a free-living organism reveals size bias in AlphaFold3 interface scores

To characterize the performance of the pooled-PPI approach, we used it to predict all pairwise PPIs in the free-living bacterium *M. genitalium* (476 proteins, 113,050 unique protein pairs, Dataset EV2) with AlphaFold3. Each of the 2027 jobs contained a random selection of 4–23 proteins (a pool) whose sizes sum to almost 5000 amino acids (median $n = 13$, Fig. EV2; Dataset EV2, "Methods"), and together queried a total of 160,772 total pairwise PPIs (some protein pairs appear in more than one pool) as well as 636,516 unique tripartite PPIs (3.56% of 17,861,900 such interactions). These pools took 68 person-days to run using the free allotment of jobs on the AlphaFold3 web interface.

AlphaFold3 evaluates the accuracy of predicted complexes using several metrics, with the interface predicted template modeling (ipTM) score (Abramson et al, 2024; Evans et al, 2021), being the most commonly used. Generally, ipTM values >0.8 are considered confident predictions, those between 0.6 and 0.8 are a gray area, and those <0.6 are incorrect predictions (Abramson et al, 2024; Evans et al, 2021). Consistent with the expectation that protein interactions are rare, very few protein pairs had high ipTM scores: 98.9% were between 0 and 0.2 (Fig. 2A).

The relative paucity of protein pairs with ipTM ≈0.0 and substantial width of the ipTM distribution (Fig. 2A) for putatively non-interacting pairs was surprising, considering that many of those protein pairs were modeled >30 Å apart. This observation led us to wonder whether some aspect of protein pairs other than their interface may influence their ipTM. Analysis revealed a strong and significant correlation between ipTM and the square root of the summed size of the interacting proteins (robust $R^2 = 0.983$, Fig. 2B, "Methods"). Since there is no apparent biological reason for this size–ipTM relationship, we predicted ipTM for each pair based on the sum of the protein sizes and subtracted this prediction from all of our ipTM values to generate size-corrected ipTM scores ("Methods", Appendix, Appendix Fig. S1). Size-corrected ipTM exhibited a much narrower distribution of scores for non-interacting protein pairs (Fig. 2A), and (as discussed below) were more predictive of biologically relevant interactions. A similar size bias is also apparent in published AlphaFold-multimer datasets (Fig. EV3A,B) and in our AlphaFold3 predictions of protein pairs (Fig. EV3C), suggesting that this bias is not due to our pooled approach but represents a more general (and potentially undesirable) feature of ipTM scores.

## Pooled-AlphaFold3 is reproducible

To assess the reproducibility of pooled ipTM measurements, we first analyzed the 38,718 protein pairs that were present in 2 or more pools. The size-corrected ipTM scores of these 38,718 proteins were well correlated in the different pools (Pearson's $r = 0.670$, Fig. 2C), especially for pairs with a size-corrected ipTM >0.4, suggesting minimal context dependence. Performing the same analysis using raw ipTM scores gave the illusion of better

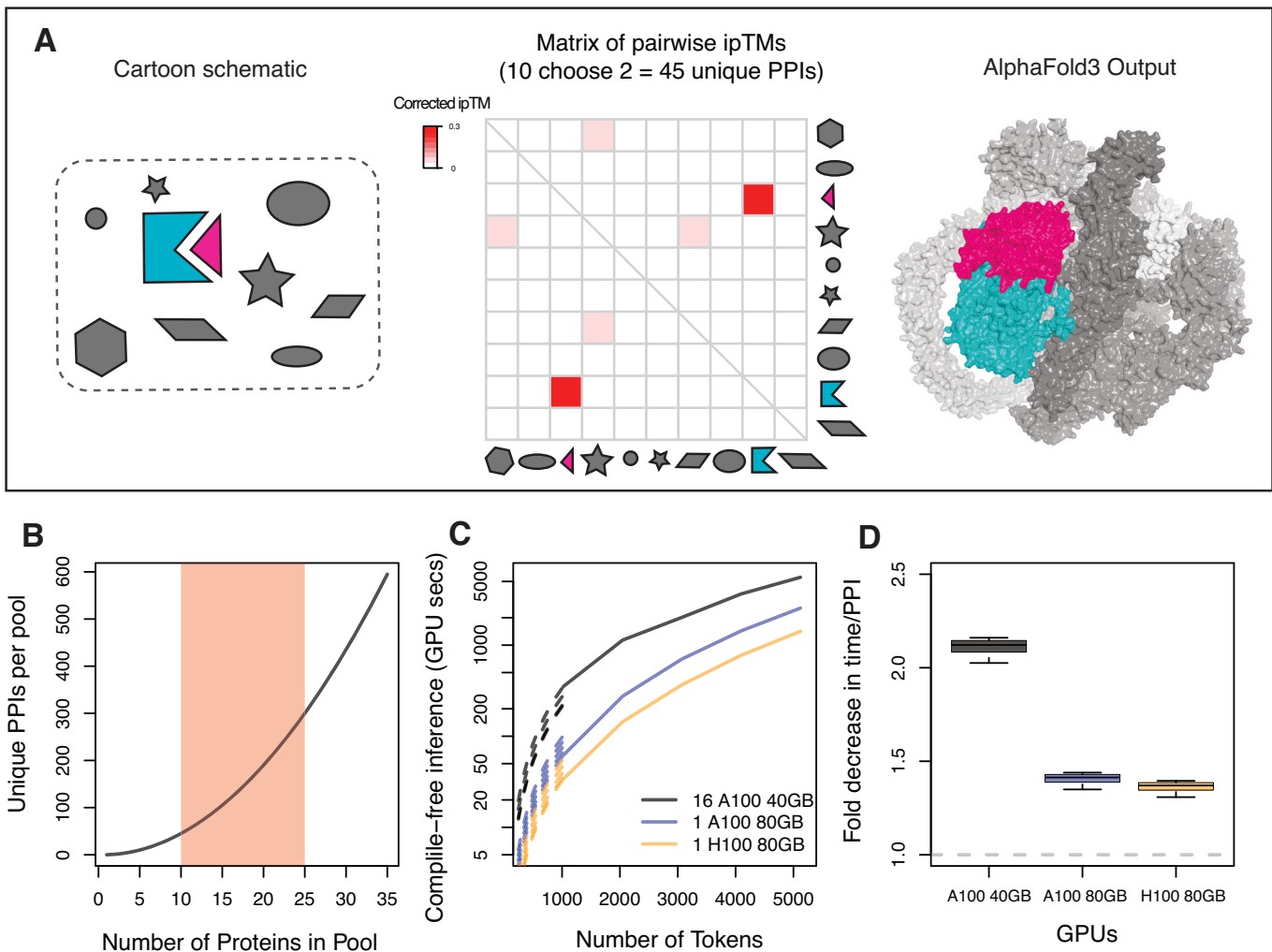

**Figure 1. Pooled-PPI prediction reduces the number of jobs and increases inference throughput.**

(A) Schematic describing pooled-PPI. A pooled job run for this study (250310_mgen_allbyall_627, described in Dataset EV2) containing 10 proteins produces a matrix of size-corrected pairwise ipTMs. 45 unique potential interactions are assayed in this job. Pooled-AlphaFold3 correctly identifies an interacting protein pair in the milieu of non-interacting pairs (MG_290 (blue) and MG_291 (red), ABC-transporter components). (B) The number of proteins in the pool determines the number of unique pairwise interactions screened in a single job. The red shaded area encompasses most jobs. (C) AlphaFold3 compile-free inference time increases as the job size increases in a GPU-dependent function. Broken lines indicate estimates from each published timing (1024–5120 tokens) based on the assumption of quadratic scaling, since the runtime for jobs <1024 tokens is not published. (D) Pooled-PPI-prediction decreases inference time per unique PPI, even considering the increased runtime of larger jobs. All timing estimates (1024–5120 tokens) and pool sizes of 10–25 proteins are considered and reflected in the error bars. Box bounds: first and third quartile, midline: median, whiskers: most extreme datapoints.

reproducibility (Pearson's $r = 0.83$, Fig. EV4, Dataset EV2), because the effect of size on raw ipTM is the same for a given protein pair. These data validate our assumption that pool context does not strongly affect the outcome of PPI predictions.

## *M. genitalium* predicted PPIs recapitulate known interactions

Having validated the technical aspects of our screen, we next tested how well the predicted PPIs captured known biological interactions. *Mycoplasma* species (like *M. genitalium*) are ideal test cases: they have streamlined genomes and have been well-studied from clinical (Viz-Lasheras et al, 2025), synthetic (Gibson et al, 2008), and basic (O'Reilly et al, 2020; Glass et al, 2006) biology

perspectives. We benchmarked our *M. genitalium* predicted-PPI dataset against the STRING database "experimental" channel, which consists of protein–protein association evidence imported from repositories such as BioGRID, PDB, and IntAct rigorously propagated through bacterial phylogeny to appropriately integrate information from related species (Szklarczyk et al, 2023). As expected, considering the known rarity of PPIs (Venkatesan et al, 2009), the vast majority (105,338/113,050, 93.2%) of protein pairs in *M. genitalium* had no evidence of interaction in the STRING experimental channel. Only 2633 protein pairs (2.3%) had strong evidence of interaction (score >800), and even fewer (1487, 1.3%) had the strongest evidence of interaction (score = 999). The average number of strong interactions per protein (5.53 in *M. genitalium*) is consistent with that of essential genes in well-studied bacteria (e.g.,

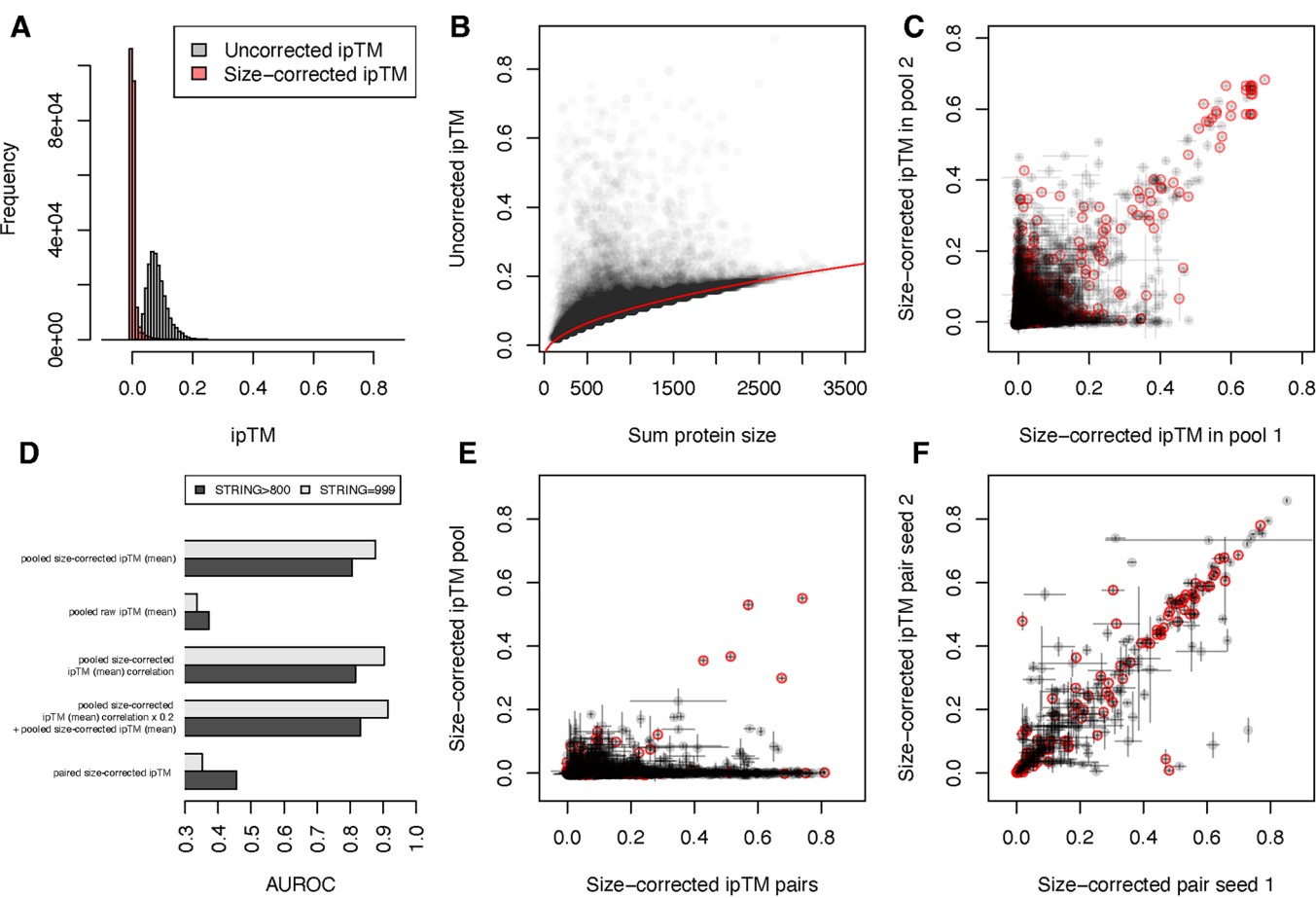

**Figure 2.  Technical characterization of pooled-AlphaFold3 ipTMs (the average of 5 diffusion samples per seed).**

(C, E, F) Circles in red are interactions with scores >800 in the experimental channel of the STRING database, and error bars represent the standard deviation of 5 diffusion samples per seed. (A) Distribution of raw (gray) and corrected (red) pairwise ipTM scores for all 113,050 unique protein pairs in the *M. genitalium* genome. (B) The ipTM for non-interacting protein pairs is proportional to the square root of the summed size of the proteins. The red fit line is sqrt(sum_protein_size)*0.0044−0.036. (C) Pooled-AlphaFold3 scores are reproducible. Size-corrected ipTM scores for the 38,718 protein pairs that appear in multiple pools are similar (Pearson's $r = 0.670$, 38,718 protein pairs). (D) Pooled-AlphaFold3 accurately predicts known interactions in the STRING database. AUROC scores for different variations compared to the STRING experimental database. (E) Pooled-AlphaFold3 scores are poorly correlated with paired approaches. Size-corrected ipTM differs between paired and pooled AlphaFold3 (Pearson's $r = 0.156$, 4560 protein pairs). (F) AlphaFold3 has intrinsic variability in ipTM scores. Size-corrected ipTM scores across multiple random seeds for runs with two proteins exhibit surprising variability (Pearson's $r = 0.878$, 314 pairs).

4.12 strong interactions per essential protein in *Bacillus subtilis*), suggesting that STRING adequately captures known PPIs in *M. genitalium*. Size-corrected ipTM scores were predictive of these known interactions, as quantified by the area under the receiver operating characteristic curves (AUROC), exhibiting an AUROC of 0.81 for strong interactions and 0.88 for the strongest interactions (Figs. 2D and EV5; Dataset EV4). Using the raw (not size corrected) ipTM values performed significantly worse (Fig. EV5, AUROC = 0.37 and 0.34 for strong and the strongest interactions, respectively), supporting our correction of this bias and raising the possibility that applying this correction could reveal new biologically relevant interactions in other datasets. The worse than random AUROC of raw ipTMs is due to the fact that interacting protein pairs are generally shorter than random pairs (median protein pair lengths: strong evidence of interaction 346 aa, random 655 aa). Strikingly, we found that the predictive power of our size-corrected ipTM scores was not limited solely to the strongest

interactions (Dataset EV4): protein pairs with size-corrected ipTM 0.1–0.2 were also significantly enriched in strong (5.8-fold) and the strongest (8.2-fold) STRING experimental interactions, as were protein pairs with size-corrected ipTM 0.05–0.1 (4.8- and 6.4-fold, respectively).

We also tested whether correlations between the predicted PPI profiles of different proteins can indicate additional functional relationships, as would be the case if two otherwise non-interacting proteins both interact with the same partner(s). Consistent with this interpretation, the correlation matrix of predicted PPIs (Dataset EV4) was also predictive of both strong (AUROC = 0.82) and the strongest (AUROC = 0.90) interactions in the STRING experimental database (Figs. 2D and EV5). While protein pairs with high size-corrected ipTM were more likely to have strongly correlated partners, numerous protein pairs without strong evidence of interaction also exhibited correlated PPI profiles (Fig. EV6A), suggesting that the correlation of PPI profiles

represents a partially orthogonal source of information that can potentially overcome defects in PPI prediction. To test this hypothesis, we combined the size-corrected ipTM matrix with the correlation matrix in varying proportions and assessed the performance of these combined scores in predicting interactions in the STRING experimental database. Combined scores slightly outperformed both individual metrics (Fig. 2D); a combined score consisting of size-corrected ipTM + 0.2 * correlation achieved good predictions for both strong (AUROC = 0.83) and for the strongest interactions (AUROC = 0.92, Figs. 2D, EV4, and EV6B). Together, these data demonstrate that pooled-AlphaFold3 sensitively and specifically captures known PPIs.

## Pooled-AlphaFold3 outperforms a paired approach for predicting PPIs

Previous in silico PPI screens were performed by co-folding two proteins per job (Schmid and Walter, 2025; Yu et al, 2023; Yirmiya et al, 2025; Burke et al, 2023; Schweke et al, 2024). To compare pairwise predictions to our pooled approach, we randomly selected 96 proteins (~20% of the *M. genitalium* proteome), used AlphaFold3 to individually co-fold all possible pairs of these proteins (4560 pairs, "Methods", Dataset EV3), and ascertained their ability to predict known PPIs. Surprisingly, the size-corrected ipTM of individually co-folded pairs was substantially worse than our pooled approach at identifying both strong (AUROC = 0.46 compared to 0.81) and the strongest (AUROC = 0.35 compared to 0.88) interactions in the STRING experimental database (Fig. 2D; Dataset EV3). Consistent with this, size-corrected ipTMs of protein pairs were poorly correlated between the paired and pooled approaches (Pearson's $r = 0.157$), despite similar per-chain pTMs (Fig. EV7A). The discrepancy was driven primarily by numerous protein pairs with significant size-corrected ipTM when folded individually and near-zero size-corrected ipTM in pools (Fig. 2E). We additionally individually co-folded 942 protein pairs covering most of the high size-corrected ipTM interactions identified by the pooled approach. This set of protein pairs exhibited highly correlated size-corrected ipTM between the pooled and paired contexts (Pearson's $r = 0.725$, Fig. EV7B). Together, our comparison between pooled-AlphaFold3 and individually folded pairs suggests a low false-positive rate for the pooled approach vis-a-vis individually co-folded protein pairs.

That pooled-AlphaFold3 outperforms a paired approach is consistent with previous work showing that "competitive" Alpha-Fold protein-peptide binding can improve predictions (Chang and Perez, 2023; Vosbein et al, 2024; Mondal et al, 2024). Since these papers used older versions of AlphaFold and tested only small numbers (i.e., 2–5) of competitive peptides per job, we next asked whether 5000 aa pools are required to achieve the full benefits of pooling. To answer this question, we randomly partitioned all *M. genitalium* proteins ($n = 476$) into five approximately equal groups, then modeled all interactions within a group using either pairs or random pools of increasing size (2000 aa, 3000 aa, 4000 aa, or 5000 aa) and assessed how well their size-corrected ipTM predicted known interactions (STRING experimental >800). Individually folded pairs had the lowest AUROC, and AUROC steadily increased as pool size increased up to 5000 aa (Figs. 3 and EV7C; Dataset EV5), suggesting that the full benefit of pooling requires at least 5000 aa pools. The increased AUROC was driven primarily by

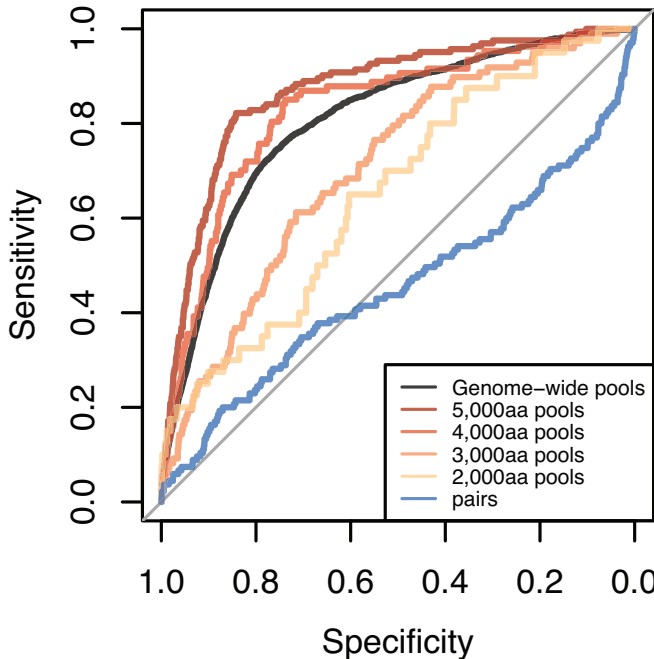

**Figure 3.  Pooled-AlphaFold3 identifies previously known PPI better than pairs.**

Plot shows ROC curves based on size-corrected ipTM for ~5000 protein pairs that were folded as pairs (blue), or in pools of increasing sizes. The largest pools exhibited the best predictive performance (as measured by AUROC). STRING experimental scores >800 (strong interactions) were used as the true-positive set.

a decrease in false-positive hits (Fig. EV7D–F). We were unable to test larger pool sizes due to a lack of GPUs with »80 GB of memory, but it is possible that larger pools could improve performance further. Our results significantly extend previous observations on the benefits of pooling (Chang and Perez, 2023; Vosbein et al, 2024; Mondal et al, 2024) and suggest that large pools with many proteins dramatically enhance the utility of genome-scale PPI predictions by reducing false-positive interactions.

## Averaging inherently variable AlphaFold3 interface scores modestly improves PPI prediction

When comparing the size-corrected ipTM across pools, much of the variability arose from protein pairs that exhibit a meaningful size-corrected ipTM (0.2–0.4) in one run, and near-zero size-corrected ipTM in another. This observation led us to ask whether the variability was due to our pooled approach or if it is inherent to the generative framework of AlphaFold3. To determine the source of the variability, we ran 314 individual pairs twice using different random seeds. Strikingly, these individually folded pairs also exhibited substantial variability in size-corrected ipTM (Pearson's $r = 0.878$, Fig. 2F). Of the 314 protein pairs that were independently assayed twice, 52 (16.6%) exhibited a difference in ipTM >0.1, 19 (6.1%) exhibited a difference in ipTM >0.2, and 8 (2.5%) exhibited a difference in ipTM >0.4, despite being identical paired (not pooled) runs.

To test whether the variability we observed (Fig. 2C) is due primarily to false-positive or false-negative errors, we evaluated the AUROC using the mean, maximum, and minimum size-corrected ipTM for the 38,718 protein pairs in multiple pools. We reasoned that if discrepant values were solely due to false positives, taking the minimum value should improve predictive performance. Similarly, if discrepant values were due solely to false negatives, taking the maximum value should improve predictive performance. Mean size-corrected ipTM was more predictive of real PPIs than either the minimum or the maximum (Dataset EV4), and the same was true for smaller pool sizes (Fig. EV7C), suggesting the variability is a combination of false-positive and false-negative errors. Averaging the size-corrected ipTM across multiple seeds modestly improves the prediction of known PPI, consistent with what has been observed for earlier versions of AlphaFold (Wallner, 2023). Together, these data suggest that variability in ipTM scores is intrinsic to AlphaFold3 and includes both false-positive and false-negative interactions.

## Pairwise interactions accurately recapitulate macromolecular complexes

Our strategy for identifying PPIs is inherently pairwise; each protein pair appears in at least one pool. However, in vivo protein interactions are rarely that simple: the functional form of many proteins is multimeric (e.g., homodimeric), and many proteins function in the context of macromolecular complexes. This prompts three related questions. First, can protein interactions in large complexes be accurately predicted by considering only pairwise interactions? Second, can structurally homologous but functionally distinct complexes be distinguished? Third, can protein–protein interfaces that are part of a larger complex be correctly predicted? To answer these questions, we considered three well-characterized higher-order complexes (the ribosome, ABC transporters, and RNA polymerase) in additional detail. Where applicable, we compared our results to those of a cross-linking mass spectroscopy (XL-MS) study performed with two different cross-linking chemicals (DSS and DSSO, 11.4 A and 10.1 A crosslinker spacer arm lengths, respectively) in a related organism, *Mycoplasma pneumoniae* (O'Reilly et al, 2020).

The ribosome is a large macromolecular complex (~2500 kDa) composed of 2 large rRNA molecules and ~50 ribosomal proteins. Most ribosomal proteins interact primarily with the rRNA, although some scaffold each other's interactions. To assess the performance of pooled-AlphaFold3 at predicting these interactions in the absence of the scaffolding 16S and 23S rRNA (which were not included in our predictions), we compared the size-corrected ipTMs of all ribosomal protein interactions with the minimum distances between ribosomal proteins extracted from a recent cryo-EM structure of the *M. pneumoniae* ribosome (Xue et al, 2022). Size-corrected ipTM was largely consistent with the minimum distances extracted from the *M. pneumoniae* ribosome structure (Xue et al, 2022) (Fig. 4A,B). Protein pairs closer than 5 Å exhibited significantly higher ipTMs than those located further apart (Fig. EV8; $t$ test $P = 1.625\text{e-}17$). Notably, pooled-AlphaFold3 was significantly more accurate at recapitulating protein–protein interactions in the ribosome than XL-MS (O'Reilly et al, 2020) (Figs. 4C and EV8), despite the high expression and stable interactions of these proteins. The ability of pooled-AlphaFold3

prediction to correctly "assemble" the ribosome in the absence of the rRNA scaffold may be due to its efficient use of structural templates (Abramson et al, 2024): although the structure (Xue et al, 2022) we used for benchmarking was not part of the default template set used, ribosome structures from other species likely served as accurate templates for ribosomal protein structures and interactions, allowing accurate PPI prediction in the absence of biological context.

*M. genitalium* encodes eight ABC-type nutrient transporters. ABC transporters are structurally similar (Fig. 4D), consisting of 2 transmembrane (TM) subunits, 2 nucleotide-binding domains (NBD), and (optionally) a periplasmic binding protein (PBP) (Davidson and Chen, 2004), and are frequently encoded in operons. We took advantage of this organization to assess the performance of pooled-AlphaFold3 in correctly assembling a large number of distinct but structurally homologous complexes. For each of the eight ABC transporters, we compared the size-corrected ipTM of biologically relevant interactions (PBP-TM, TM-TM, and TM-NBD) both within a single transporter and between different transporters. All three types of interactions exhibited significantly higher size-corrected ipTMs within a single transporter than between transporters (Fig. 4E), and individual transporters were clearly discernible in an unclustered heatmap of pairwise size-corrected ipTMs, with very little crosstalk (Fig. 4E). Pooled-AlphaFold3 discerned ABC transporters better than XL-MS (Fig. 4F), highlighting the independence of our in silico predictions from protein abundance. Our analysis of ABC transporters strongly supports the idea that pooled-AlphaFold3 can sensitively and specifically segregate and assemble structurally homologous complexes.

RNA polymerase is a multisubunit enzyme composed of $\alpha\alpha\beta'\beta\delta\sigma^A$ and functions as a key regulatory node. We found strong predicted interactions between the 5 subunits of RNA polymerase (RpoABCDE), GreA (a small protein that interacts with the secondary channel to rescue stalled RNA polymerases (Borukhov et al, 1992; Opalka et al, 2003; Abdelkareem et al, 2019)), MG_354, a DUF1951 protein previously shown to interact with RNA polymerase (O'Reilly et al, 2020), and other RNA polymerase interacting proteins such as NusG (Kang et al, 2018), UvrD (Kawale and Burmann, 2020), and Spx (Shi et al, 2021) (Fig. 5; Dataset EV4). Since many of these interactions have been structurally characterized, we asked whether our largely pairwise interactions accurately recapitulated the structure of interactions in and with this complex. We compared each pairwise structure of RNAP subunits (extracted from pools of other proteins) to a predicted structure of the complete *M. genitalium* holoenzyme ($\alpha\alpha\beta'\beta\delta\sigma^A$, generated using AlphaFold3; Fig. 5) or experimental structures of relevant complexes (see Table EV1).

Pairwise predicted interfaces closely matched the same interfaces in the holoenzyme. For instance, a job run for this study, 250310_mgen_allbyall_5, co-folded 17 *M. genitalium* proteins, including three RNAP subunits, αII, β', and δ (Table EV1), which were correctly predicted to interact (Dataset EV2). The αII-β'-δ complex predicted from 250310_mgen_allbyall_5 superimposed on the same subunits of the *M. genitalium* RNAP with an RMSD of 0.845 Å over 1,091 a-carbons (using the PyMOL align command to exclude flexible loops that do not align well). Predicted interactions between RNAP subunits and other RNAP-interacting proteins GreA, NusG, and Spx were compared with available experimental

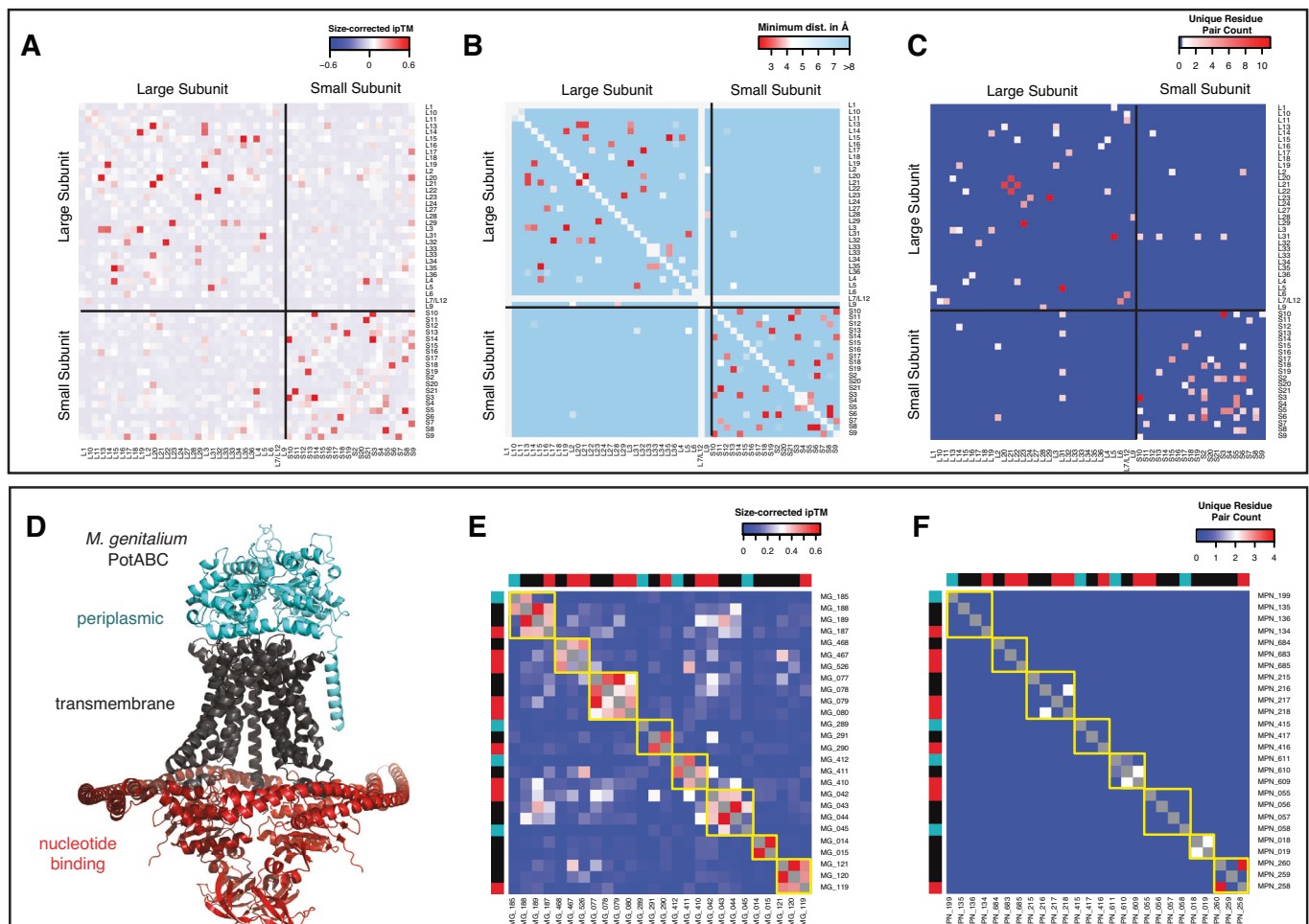

**Figure 4. Pairwise interactions recapitulate macromolecular complexes.**

(A–C) Colors are such that nearby protein pairs should be in red, leading to a similar pattern of red dots in all three panels. (E, F) ABC transporter domains (periplasmic, transmembrane, nucleotide-binding) are color-coded in the row and column sides consistent with (D), and interactions within individual ABC transporters are outlined in yellow. (A) Size-corrected ipTM values between all ribosomal proteins in *M. genitalium* from the pooled-AlphaFold3 screen. Red denotes high size-corrected ipTM values. Note that almost all high-size-corrected ipTM interactions occur between protein pairs in the same ribosomal subunit. (B) Minimum distance between ribosomal proteins in the published cryo-EM structure of the *M. pneumoniae* ribosome (Xue et al, 2022). Red denotes protein pairs close together. (C) Unique residue pairs between ribosomal proteins in published *M. pneumoniae* XL-MS data. Red denotes protein pairs with identified cross-links, indicating proximity. (D) Predicted AlphaFold3 complex of the *M. genitalium* ABC transporter PotABC with domains labeled and color-coded. (E) Size-corrected ipTMs between the eight ABC-type nutrient transporters. (F) Unique residue pairs between ABC-type transporters in *M. pneumoniae* XL-MS data (O'Reilly et al, 2020).

structures (Table EV1). Overall, 16 jobs included at least one RNAP subunit and another interacting protein (Table EV1) and the predicted interactions superimposed with the corresponding proteins in the *M. genitalium* RNAP holoenzyme or with the experimental structures with a mean RMSD of 0.899 ± 0.312 Å (median number of a-carbons included in alignments was 1109). As with the ribosome, the efficient use of structural templates likely played an important role in the accuracy of these predictions. These data suggest that our inherently pairwise pooled-AlphaFold3 predictions accurately predict not only the interacting partners but also the interaction interfaces (at least for RNA polymerase and other structurally characterized complexes), allowing the use of this method for structural analyses, such as the identification of competitive (or synergistic) protein binding interfaces.

## Pooled-AlphaFold3 suggests novel interactions for *M. genitalium* proteins

Given that our *M. genitalium* predicted PPIs were highly predictive of known biological interactions and complexes, we visualized the combined ipTM and correlation data (all combined scores >0.2) as a network (Fig. 6A). Genes of similar function frequently clustered together: transporters, ribosomal proteins, RNA polymerase subunits, and DNA replication/repair proteins each formed dense interconnected networks (Fig. 6A). We used this network to manually assess previously unknown interactions to determine if they could inform new hypotheses about *Mycoplasma* protein functions. Below, we (briefly) highlight several such stories present in our dataset.

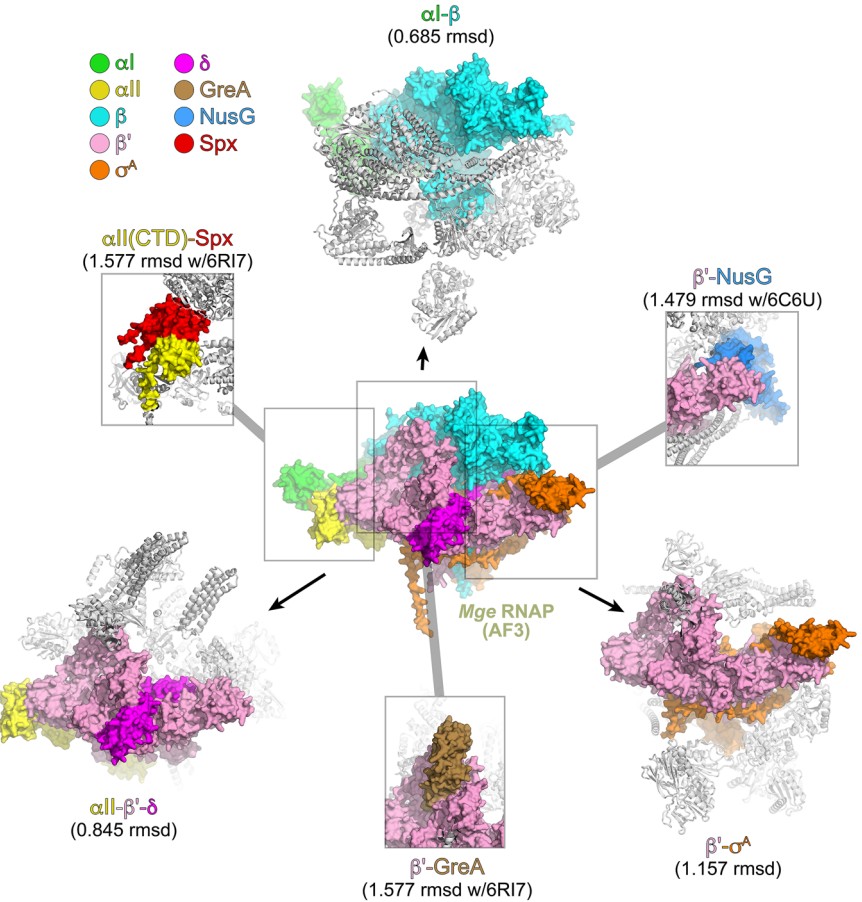

**Figure 5. Pooled-AlphaFold3 accurately predicts known interactions with RNAP subunits.**

Shown in the center is the AF3-predicted structure of *M. genitalium* (Mge) RNAP (αI-αII-β-β′-δ-σ^A); the proteins are shown as molecular surfaces and color-coded according to the legend (upper left). Shown on the top, lower right, and lower left are selected examples of pooled-AlphaFold3 predictions that included interacting RNAP subunits: (top) 250310_mgen_allbyall_1130 included αI (green molecular surface) and β (cyan molecular surface) in a milieu of eight other proteins (shown as gray ribbons). These proteins from the pool superimposed with the equivalent proteins from Mge RNAP with an RMSD of 0.685 Å (Table EV1). (lower right) 250310_mgen_allbyall_2016 included β′ (pink) and σ^A (orange) in a milieu of seven other proteins (gray ribbons). These proteins from the pool superimposed with the equivalent proteins from Mge RNAP with an RMSD of 1.157 Å (Table EV1). (lower left) 250310_mgen_allbyall_5 included αI (yellow), β′ (pink), and δ (magenta) in a milieu of 14 other proteins (gray ribbons). These proteins from the pool superimposed with the equivalent proteins from Mge RNAP with an RMSD of 0.845 Å (Table EV1). Boxed are the PPIs for predictions of other RNAP-interacting proteins with RNAP subunits for which experimental structures are available: (top right) 250310_mgen_allbyall_1290 included β′ (pink) and NusG (blue) in a milieu of 11 other proteins (gray ribbons). (bottom) 250310_mgen_allbyall_1607 included β′ (pink) and GreA (brown) in a milieu of nine other proteins (gray ribbons). (top left) 250310_mgen_allbyall_256 included (yellow) and Spx (red) in a milieu of 11 other proteins (gray ribbons). All of the "250310_mgen_allbyall_" jobs are described in Dataset EV2.

### A heterodimeric RNase Y in M. genitalium and M. pneumoniae

In Firmicutes, RNase Y is a quasi-essential RNase responsible for mRNA turnover (Benda et al, 2021). Like its functional homolog in gram-negative bacteria (RNase E), RNase Y is membrane-tethered, and membrane attachment is important for its function (Laalami et al, 2024). RNase Y typically forms a homodimer, with a coiled-coil domain between the N-terminal transmembrane helix and the catalytic C-terminal forming the principal dimerization interface (Fig. 6B) (Morellet et al, 2022). Strikingly, although AlphaFold3 correctly folded the *B. subtilis* Rny homodimer, the *M. genitalium* Rny (MG_130) homodimer folded incorrectly with a low ipTM (Fig. 6B; Dataset EV6). However, these bacteria possess a distant homolog of RNase Y (MG_123/MPN_262, ~5% identity to RNase Y, member of COG1418). Our pooled-AlphaFold3 data suggest that this protein dimerizes with RNase Y (Fig. 6B), forming a

heterodimer instead, and the predicted structure of this complex is consistent with the XL-MS data (7/9 unique cross-links between Rny and MPN_262 within 30 Å, Dataset EV7). Supporting the idea that RNase Y in these species is a heterodimer, MG_123/MPN_262 is essential, as is RNase Y (Glass et al, 2006). The functional implications of this heterodimeric RNase Y remain to be explored, but may be related to the unusual post-transcriptional regulatory mechanisms postulated for these genome-reduced bacteria (Yus et al, 2019).

### A potential role in protein secretion for an essential truncated trigger factor protein

Trigger factor (Tig) is a ribosomal and cytoplasmic chaperone with an additional role in post-translational protein secretion (De Geyter et al, 2020; Martinez-Hackert and Hendrickson, 2009; Wu et al,

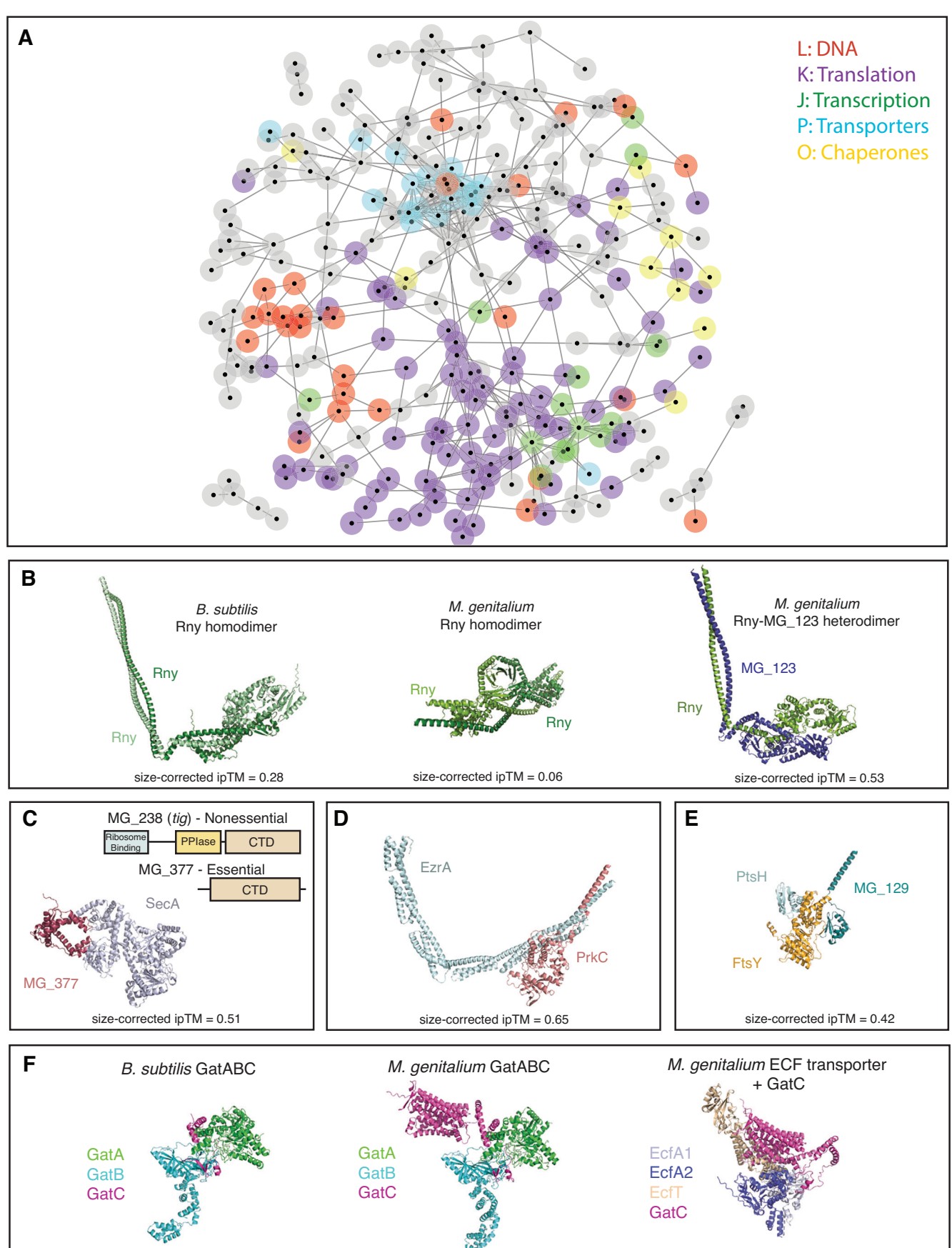

**A**

L: DNA
K: Translation
J: Transcription
P: Transporters
O: Chaperones

**B**

*B. subtilis* Rny homodimer

Rny
Rny

size-corrected ipTM = 0.28

*M. genitalium* Rny homodimer

Rny
Rny

size-corrected ipTM = 0.06

*M. genitalium* Rny-MG_123 heterodimer

MG_123
Rny

size-corrected ipTM = 0.53

**C**

MG_238 (*tig*) - Nonessential

Ribosome Binding | PPIase | CTD

MG_377 - Essential

SecA | CTD

MG_377

size-corrected ipTM = 0.51

**D**

EzrA
PrkC

size-corrected ipTM = 0.65

**E**

PtsH
MG_129
FtsY

size-corrected ipTM = 0.42

**F**

*B. subtilis* GatABC

GatA
GatB
GatC

size-corrected ipTM = 0.76

*M. genitalium* GatABC

GatA
GatB
GatC

size-corrected ipTM = 0.55

*M. genitalium* ECF transporter + GatC

EcfA1
EcfA2
EcfT
GatC

size-corrected ipTM = 0.64

**Figure 6. Novel *M. genitalium* predicted PPIs.**

(A) Network of genome-wide PPIs in *M. genitalium*. Genes are represented by nodes, and edges are set as (size-corrected ipTM + 0.2 * correlation(ipTM)). The layout is edge-weighted spring-embedded. All edges with weight >0.2 are shown. Genes in specific COG categories are highlighted by node borders: Translation (J) = purple, Transcription (K) = green, DNA (L) = red, Ion Transport (P) = cyan, Chaperones (O) = yellow. All other COGs, including Unknown (S), are shown in gray; clusters of these proteins may represent spurious interactions. Transcription (RNA polymerase) genes cluster within translation due to the presence of predicted interactions related to transcription-translation coupling (e.g., NusA, NusG, RpsB, RpsE). (B) Predicted structures of *B. subtilis* Rny homodimer, *M. genitalium* Rny homodimer exhibiting incorrect structure, and *M. genitalium* Rny/MG_123 heterodimer exhibiting correct fold. (C) Schematic of the domain organization of the two Tig homologs in *M. genitalium* and predicted interaction between SecA and the essential *tig* homolog MG_377. The interacting residues of MG_377 are located near the center of its CTD domain. (D) Predicted interaction between EzrA and the serine/threonine kinase PrkC. (E) Predicted structure of FtsY-PtsH-MG_129 complex, showing independent interfaces of two PTS components with FtsY. (F) Predicted complexes of *B. subtilis* GatABC, *M. genitalium* GatABC, and *M. genitalium* EcfA$_1$A$_2$T-GatC, showing the S-protein domain of GatC in *M. genitalium* and its predicted interaction with the ECF transporter.

2022). Tig consists of three domains: an N-terminal ribosome-binding domain, a central FKBP-type peptidyl-prolyl cis-trans isomerase chaperone domain, and a C-terminal domain involved in chaperone functions (Fig. 6C). Trigger factor is ubiquitous in bacterial genomes, non-essential, and almost never duplicated due to a severe dosage constraint caused by the ribosome-binding N-terminal domain (Wan et al, 2024). In the few genomes with multiple trigger factor-like genes, only one usually contains the N-terminal domain (Wan et al, 2024). *M. genitalium* and the related *M. pneumoniae* each encode a full-length non-essential Tig and an essential Tig homolog containing only the C-terminal chaperone domain (MG_377/MPN555, Fig. 6C) (O'Reilly et al, 2020; Glass et al, 2006). Our data suggest a role for MG_377/MPN555 in protein secretion: both *M. genitalium* MG_377 and *M. pneumoniae* MPN555, but not their canonical Tigs, have strong predicted interactions with SecA, the membrane subunit of the bacterial Sec apparatus (Datasets EV4 and EV6; Fig. 6C). This predicted interaction may be essential for SecA function due to the lack of other secretory chaperones such as SecB or CsaA (Linde et al, 2003) in *Mycoplasma*.

### A potential role for serine/threonine kinase in Mycoplasma cell division

Tenericutes, including *Mycoplasma*, lack a peptidoglycan cell wall, making their cell division process fundamentally different from that of other bacteria, where an FtsZ ring organizes septal peptidoglycan synthesis (Lluch-Senar et al, 2010). Thus, relatively little is known about cell division in *Mycoplasma*. Our dataset predicts an interaction between the cell division-associated protein EzrA/MG_397 and the eukaryotic-like serine/threonine kinase PrkC/MG_109 (Fig. 6D). AlphaFold3 predicts an interaction between these proteins in both the closely related *M. pneumoniae* and in the more distantly related Tenericute *Spiroplasma citrii* (Dataset EV6), but not between EzrA and any of the three serine/threonine kinases in the distantly related Firmicute *B. subtilis* (Dataset EV6). Deletion of PrkC in *M. pneumoniae* results in a significant decrease in EzrA protein concentration (van Noort et al, 2012), a pattern shared with the cytoskeletal HMW1-5 proteins, which require phosphorylation by PrkC for stability in *M. pneumoniae* (Schmidl et al, 2010). Intriguingly, the spectrin-like domains of EzrA exhibit a coiled-coil motif, as do large portions of the HMW1-5 proteins (Balish, 2014) and *B. subtilis* GpsB (a target of PrkC in that species (Pompeo et al, 2015)), raising the possibility that PrkC preferentially recognizes and phosphorylates coiled-coils to regulate cell division.

### Moonlighting enzymes in essential processes

*M. genitalium* and *M. pneumoniae* are the products of extreme genome reduction, which has been associated with neofunctionalization of proteins leading to moonlighting enzymes (Kelkar and Ochman, 2013). Below, we discuss two novel examples of moonlighting revealed in our data.

FtsY is involved in SRP-mediated co-translational protein targeting to the membrane (Miller et al, 1994). We were surprised to find that our pooled-AlphaFold3 screen identified two proteins involved in the phosphoenolpyruvate-dependent sugar phosphotransferase system (PTS) as interactors of FtsY: MG_129, an orphan PTS EIIB protein (size corrected ipTM = 0.496), and MG_041, the phosphocarrier protein PtsH (size corrected ipTM = 0.317). A single crosslink between FtsY and MPN_268/MG_129 was identified in the XL-MS study (O'Reilly et al, 2020), which connected residues 33.4 Å apart in the predicted complex (Fig. 6E; Dataset EV7). No single EIIC or EIIA components are encoded in the *M. genitalium* genome, supporting the idea that MG_129 moonlights in a different process, such as co-translational protein secretion.

ECF transporters consist of EcfA1, EcfA2, and EcfT components, which combine with a specificity protein (S) to allow the import of various micronutrients (Rempel et al, 2019). Strikingly, both GatC and GatB, components of the essential Glu-tRNA$^{Gln}$ amidotransferase (GatABC) (Sheppard and Söll, 2008), were predicted to interact with ECF transporter components. The heterotrimeric enzyme GatABC transamidates mis-acylated Glu-tRNA$^{Gln}$, thereby functionally replacing glutaminyl-tRNA synthetase (which is not found in Gram-positive bacteria such as *M. genitalium*). Analysis of the predicted structure of *M. genitalium* GatC revealed an additional ~300 aa N-terminal domain with structural homology to S-proteins which is not present in GatC proteins outside of Mycoplasma and Ureaplasma and is predicted to bind to the same spot on the ECF transporter as an S-protein (Fig. 6F). The interaction between GatC and the ECF transporter is supported by the *M. pneumoniae* XL-MS data (O'Reilly et al, 2020), which identifies cross-links between GatC and several components of the ECF transporter consistent with our predicted structure (3/3 cross-links between GatC and ECF transporter components within 30 Å; Dataset EV7). The functional implications of this interaction remain to be explored: the S-protein domain of GatC may moonlight in the import of a substrate for its transamidase function, may import a completely unrelated metabolite, or may affect GatABC function via membrane anchoring.

# Discussion

Genome-scale PPI prediction is a powerful strategy for revealing new biology, but its broad adoption has been hampered by its high computational demands and a significant false-positive rate. Pooled-PPI prediction increases the accuracy of genome-scale approaches while decreasing the overall runtime as well as the number of discrete jobs of such screens and can even allow some genome-scale queries to be run on free web-based resources, democratizing access to this powerful tool. We demonstrate the utility of this strategy by predicting all pairwise PPIs (113,050) in the *M. genitalium* genome using only 2027 jobs, which required just 68 person-days using our personal AlphaFold3 accounts. Our results reveal a wealth of technical insights into AlphaFold3 as well as biological insights into *M. genitalium* RNA metabolism, protein secretion, and cell division that will drive future experimental studies.

Our study assessed all possible pairwise interactions in an organism (rather than carefully curated subsets of known negative and positive interactions), and ~1/3 of interactions were assayed more than once (by default), allowing us to uncover several important features of AlphaFold3. First, our study revealed that pooled-AlphaFold3 significantly outperforms paired AlphaFold3 for identifying known PPIs, by largely eliminating false-positive interactions through a "decoy" mechanism. Second, the preponderance of non-interacting protein pairs in our data enabled us to visualize and correct for the size dependence of ipTM scores. Although several shortcomings of ipTM scores have been previously described (Kim et al, 2024; Dunbrack, 2025; Varga et al, 2025), the size dependence of non-interacting ipTM scores has not been explicitly noted. Correcting for this bias greatly increased the performance of ipTM in predicting known interactions and is likely to be applicable to other studies as well. Third, our dataset revealed the variability of AlphaFold3 ipTM scores in both pooled and paired settings. This variability has likely remained unappreciated, both because the size bias of ipTM scores obfuscates discrepancies and because multiple seeds are seldom run due to the high computational costs. Finally, our in-depth analysis of predicted PPIs between ribosomal proteins, RNA-polymerase subunits, and ABC-transporters revealed that AlphaFold3 does not generally require fully functional biological complexes to accurately predict interacting proteins and their interfaces, or to distinguish structurally homologous (but functionally distinct) complexes.

Pooled-PPI prediction usefully identifies known and biologically relevant PPIs, and can (in limited cases) enable the use of web-interfaces to perform genome-scale PPI prediction. It can be used to query all pairwise interactions between a set of proteins (all-by-all), or (less efficiently) for few-by-all screens. Facile access to highly specific genome-scale PPI screens has major implications. First, combining genome-wide PPI information with high-dimensional phenotypic screens, such as chemical genomics, can uncover mechanistic insights into functional protein–protein interactions, greatly accelerating our understanding of biological processes by highlighting novel connections that may not be credible in the absence of orthogonal data. Second, future pooled-PPI prediction screens could also incorporate RNAs and other molecules to assay their interacting partners: such studies could uncover novel tRNA-, rRNA- and/or ncRNA-interacting proteins.

Third, genome-scale PPI screens can be used to probe the biology of uncultured organisms, such as the Asgard archaea, which contain the relatives of the last eukaryotic common ancestor. Finally, PPI screens can be used to study intractable or fleeting interactions between organisms, such as between hosts and pathogens, members of ecological communities, or communities and hosts (e.g., the microbiome), potentially revealing novel protein determinants and drug targets. Pooled-PPI prediction democratizes the ability to make genome-scale PPI predictions (whether all-by-all or selected subsets), allowing more labs to apply this tool to more problems and ultimately furthering our understanding of biology.

# Methods

**Reagents and tools table**

| Reagent/resource | Reference or source | Identifier or catalog number |
|---|---|---|
| **Software** | | |
| R v4.3.3 | https://www.r-project.org | |
| AlphaFold3 (3.0.1) | https://github.com/google-deepmind/alphafold3 | |
| AlphaFold3 Web Server (3.0.1) | https://alphafoldserver.com/ | |
| Our pooling code | https://github.com/horiatodor/pooled-af3 | |

## Preparing pools

The genome of *M. genitalium* was downloaded from NCBI (GenBank: L43967.2, Dataset EV1). Proteins were pooled to minimize the number of pools required to co-fold all protein pairs at least once while keeping all jobs <5,000 tokens. This was accomplished using a greedy algorithm available at https://github.com/horiatodor/pooled-af3. Briefly, pools are initially completely random, but become increasingly biased for including proteins that have the largest number of missing interactions.

For the random paired, 2k, 3k, 4k, and 5k pool samples, the genome was first divided into 5 approximately equal sections (~96 proteins each). All interactions between proteins in each section were then assayed using random pools to cover all possible pairs.

## Running AlphaFold3

All 2027 comprehensive pools as well as some paired jobs were successfully run using default settings on the AlphaFold3 server (https://alphafoldserver.com/). Additional paired and pooled jobs were run locally using AlphaFold 3.0.1 and default settings (1 seed, 5 diffusion samples, 10 recycles, cutoff date of September 30, 2021 for templates).

## Analysis of AlphaFold3 runs

All AlphaFold3 runs (pooled or pairs) were downloaded. Raw ipTM scores were extracted from the "summary_confidences" files for all

5 models and averaged. The ipTM of protein pairs that appeared in multiple pools was averaged across all pools unless otherwise noted. Analysis code is available at https://github.com/horiatodor/pooled-af3.

### Size correction of ipTM scores

To determine the effect of summed protein size on ipTM, we performed a robust linear regression using the robustbase::lmrob function, which computes a MM-type regression estimator (a robust and efficient estimator with a ~50% breakdown point and 95% efficiency). A robust estimator was used to mitigate the influence of true-positive interactions on the regression. The linear regression was performed on the square root of the summed size of the two proteins (in amino acids) to calculate an "expected ipTM" for each protein pair. For the *M. genitalium* dataset, expected_ipTM = $-0.036255571 + 0.004470512*\mathrm{sqrt}(\mathrm{aa\_in\_protein1} + \mathrm{aa\_in\_protein2})$. We expect these coefficients to be similar for other datasets. Expected ipTM was subtracted from the observed ipTM to generate the "size-corrected ipTM". Code for implementing this correction is available at https://github.com/horiatodor/pooled-af3."

### STRING data

STRING data related to taxid: 243273 was downloaded on 3/18/2025 from https://string-db.org/. Only the experimental channel was used for benchmarking PPIs.

### Structural analysis

For all of the structural analyses, we focused on the top-scoring model. The analysis of RNAP is described in Table EV1. For the ribosome, mmCIF files were loaded into R and manipulated using the Bio3D package (Grant et al, 2021). Distances and RMSD were computed using the functions in this package. The structure of the *M. pneumoniae* ribosome (Xue et al, 2022) (7P6Z) downloaded from PDB was used.

### Cross-linking analysis

XL-MS data (O'Reilly et al, 2020) was mapped onto AlphaFold3 models of relevant *M. pneumoniae* protein complexes using PyXlinkViewer (Schiffrin et al, 2020). Satisfaction threshold was set to 30 Å. Both "internal" and "between" cross-links were mapped for all proteins within the complex.

## Data availability

All data derived from alphafoldserver.com was deposited under the original AlphaFold3 Server Output License in Zenodo: https://doi.org/10.5281/zenodo.15499631. All locally generated AlphaFold3 data were deposited under the AlphaFold3 license in Zenodo: https://zenodo.org/records/16920556.

The source data of this paper are collected in the following database record: biostudies:S-SCDT-10_1038-S44320-026-00189-7.

## Peer review information

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

## Acknowledgements

We thank Google DeepMind and Isomorphic Labs for providing free online access to AlphaFold3 to us and to the greater research community. We thank T Kortemme, T Goddard, D Todor, J Rock, D Booth, A Johnson, M Garber, S Silas, A Typas, and members of the Gross and Beltrao Labs for extensive helpful discussions. SAD was supported by National Institutes of Health grant R35 GM118130. PB was supported by Helmut Horten Stiftung and the ETH Zurich Foundation. CAG was supported by National Institutes of Health grant R35 GM118061.

## Author contributions

**Horia Todor**: Conceptualization; Supervision; Investigation; Visualization; Methodology; Writing—original draft; Project administration; Writing—review and editing. **Lili M Kim**: Conceptualization; Investigation; Visualization; Methodology; Writing—original draft; Writing—review and editing. **Jurgen Jänes**: Investigation; Visualization; Methodology; Writing—original draft;

Writing—review and editing. **Hannah N Burkhart**: Investigation; Writing—original draft; Writing—review and editing. **Seth A Darst**: Funding acquisition; Investigation; Visualization; Writing—original draft; Writing—review and editing. **Pedro Beltrao**: Conceptualization; Supervision; Funding acquisition; Methodology; Writing—original draft; Writing—review and editing. **Carol A Gross**: Conceptualization; Supervision; Funding acquisition; Methodology; Writing—original draft; Project administration; Writing—review and editing.

Source data underlying figure panels in this paper may have individual authorship assigned. Where available, figure panel/source data authorship is listed in the following database record: biostudies:S-SCDT-10_1038-S44320-026-00189-7.

## Disclosure and competing interests statement

The authors declare no competing interests.

# Expanded View Figures

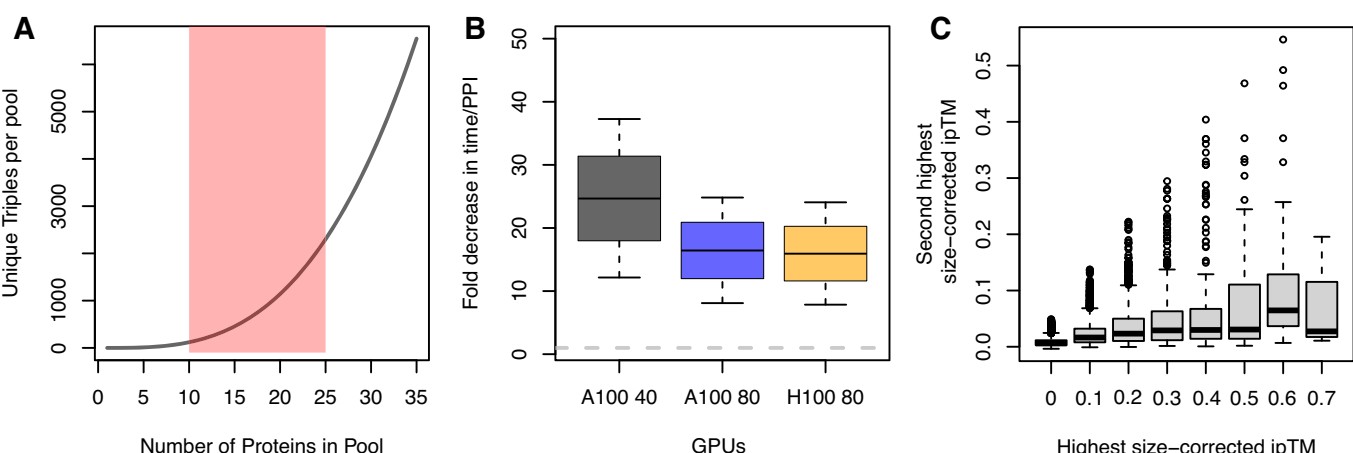

**Figure EV1.   Pooled-PPI prediction drastically reduces the number of required jobs and increases inference throughput for identifying tripartite interactions.**

(**A**) The number of proteins in the pool determines the number of unique tripartite interactions screened in a single job. The red shaded area encompasses most jobs. (**B**) Pooled-PPI-prediction decreases inference time per unique tripartite PPI even considering the increased runtime of larger jobs. All timing estimates (1024–5120 tokens) and pool sizes of 10-25 proteins are considered and reflected in the error bars. Box bounds: first and third quartile, midline: median, whiskers: most extreme datapoints. (**C**) Tripartite interactions are rare in our dataset – most proteins in most jobs have 0 or 1 strong interactions. Even in cases where a protein strongly interacts with 2 partners, identifying a true tripartite interaction would require having additional pools with each of the two partners individually and most large complexes are accurately modeled in a pairwise manner (described later in the manuscript). Plot compares the highest size-corrected ipTM to the second highest size-corrected ipTM for all proteins in each pool ($n = 25{,}985$). Box bounds: first and third quartile, midline: median, whiskers: most extreme datapoints within 1.5× interquartile range.

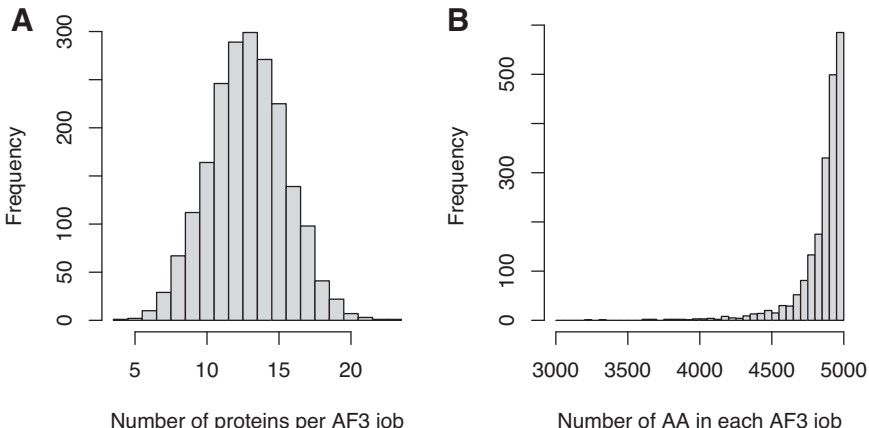

**Figure EV2.  Characteristics of the *M. genitalium* pools.**

(**A**) Histogram of pool sizes (Datasets EV1 and EV2). (**B**) Histogram of total job size (in amino acids, Datasets EV1 and EV2).

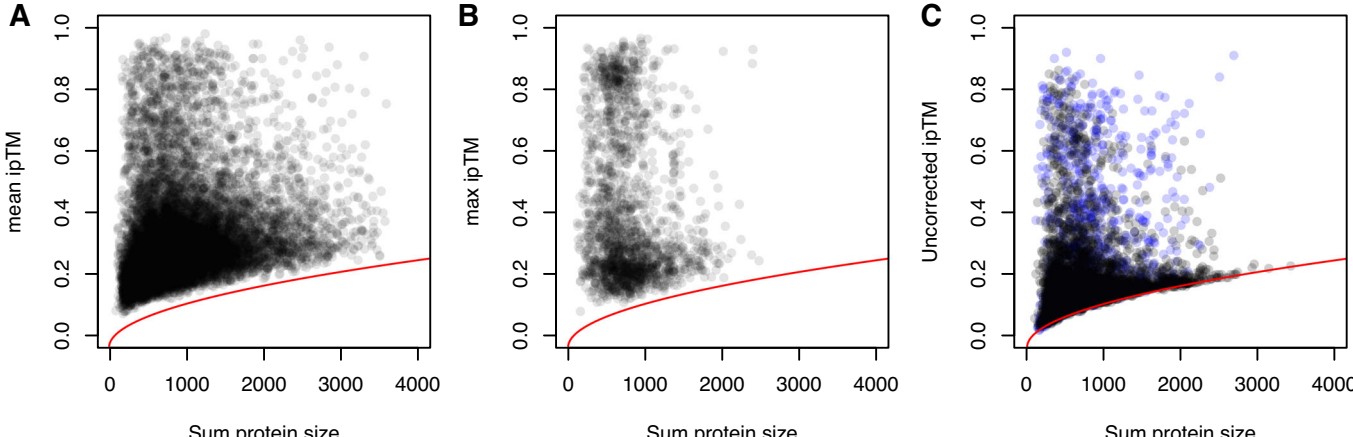

**Figure EV3. Size-bias of ipTM scores is not limited to pooled approaches or to AlphaFold3.**

Red line in all panels is the fit line from AlphaFold3 pools: sqrt(sum_protein_size)*0.0044–0.036. (A) Summed protein size plotted by ipTM for ColabFold v1.5.2/ AlphaFold-Multimer from (Schmid and Walter, 2025) exhibits a similar pattern to our data, albeit a lower correlation (robust $R^2 = 0.162$, $n = 13,274$), likely due to a lower proportion of non-interacting protein pairs. (B) Summed protein size plotted by ipTM for AlphaFold-Multimer v2.1.0 data from (O'Reilly et al, 2023) exhibits a similar pattern to our data, albeit an even lower correlation (robust $R^2 = 0.021$, $n = 1977$), likely due to a very low proportion of non-interacting protein pairs. (C) Summed protein size plotted by ipTM for individual pairs folded by AlphaFold3 exhibits a similar pattern albeit a lower correlation. Two sets of data are shown. (1) Protein pairs selected primarily from high-scoring interactions in pools and run on alphafoldserver.com (robust $R^2 = 0.170$, $n = 942$, blue). (2) Random sample of 4560 protein pairs run using a local implementation of AlphaFold3 (robust $R^2 = 0.236$, $n = 4560$, black).

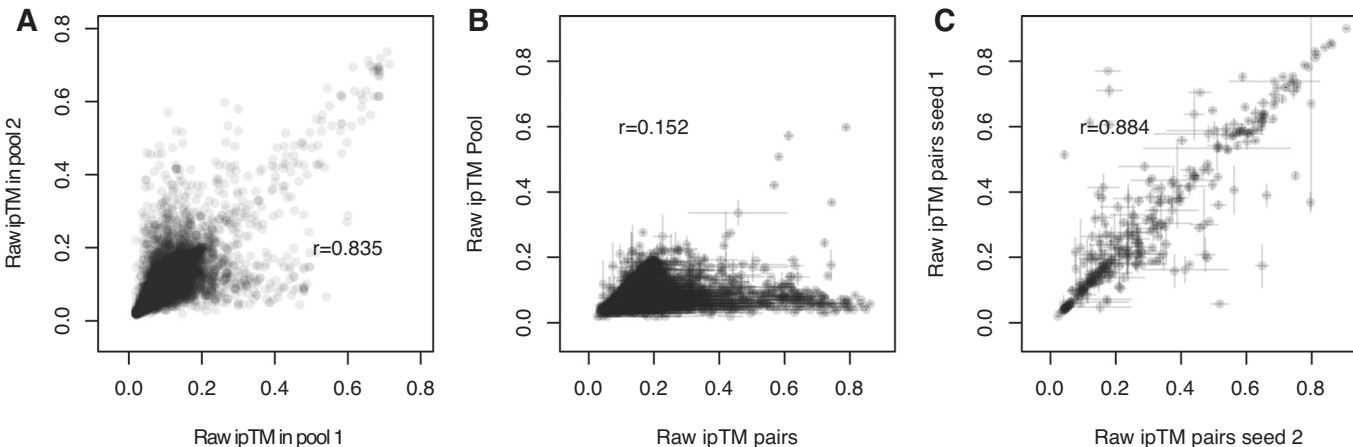

**Figure EV4.  ipTM-size bias camouflages variability in AlphaFold3.**

(A) Raw ipTM scores for the 38,718 protein pairs that appear in multiple pools are similar (Pearson's $r = 0.835$, 38,718 protein pairs). (B) Raw ipTM is similar in paired and pooled AlphaFold3 jobs (Pearson's $r = 0.152$, 4560 pairs). Error bars represent the standard deviation of 5 diffusion samples per seed. (C) Raw ipTM scores across identical paired runs using different random seeds exhibit surprising variability (Pearson's $r = 0.884$, 314 pairs). Error bars represent the standard deviation of 5 diffusion samples per seed.

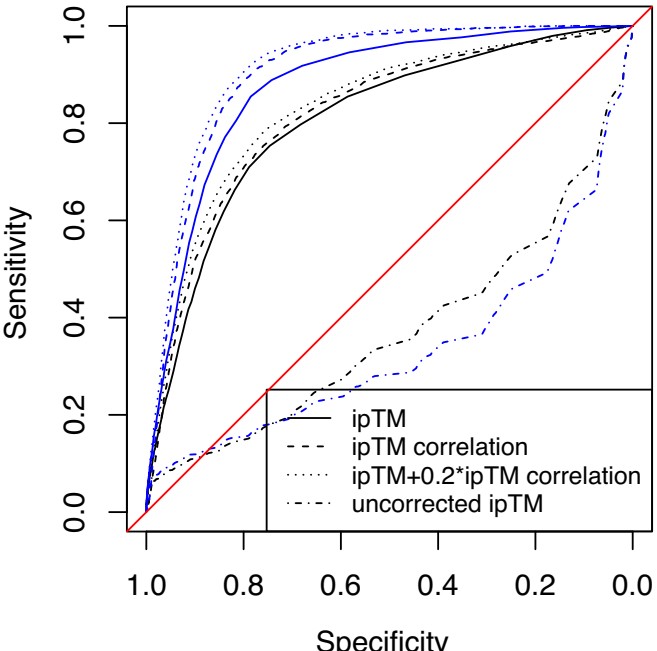

**Figure EV5.  Pooled-AlphaFold3 accurately predicts known interactions in the STRING database.**

AUROC curve for size-corrected ipTM scores (solid line), size-corrected ipTM correlations (large dashed line), size-corrected ipTM + 0.2 size-corrected ipTM correlation (small dashed line), and uncorrected ipTM (alternating small and large dashed line). Black lines consider interactions with STRING experimental scores >800 (strong interactions) as the true-positive set. Blue lines consider interactions with STRING experimental scores = 999 (strongest interactions) as the true-positive set.

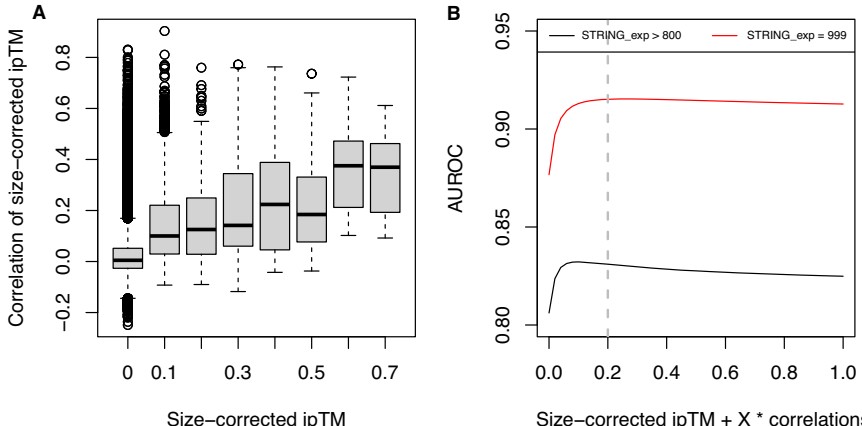

**Figure EV6. Size-corrected ipTM and the correlation between the size-corrected ipTM of two proteins are partially orthogonal, and combining the two increases predictive performance.**

(A) Boxplot showing partial correlation between the size-corrected ipTM of two proteins and the correlation between their size-corrected ipTM profiles. (B) AUROC benchmarking the STRING experimental dataset with different combinations of size-corrected ipTM and its correlation. For STRING > 800 (black line), the maximum AUROC is 0.832 and is achieved at (size-corrected ipTM + 0.1 correlation). For STRING = 999 (red line), the maximum AUROC is 0.915 and is achieved at (size-corrected ipTM + 0.26 correlation). We use a combined score of (size-corrected ipTM + 0.2 correlation) for the remainder of the manuscript (dashed vertical line).

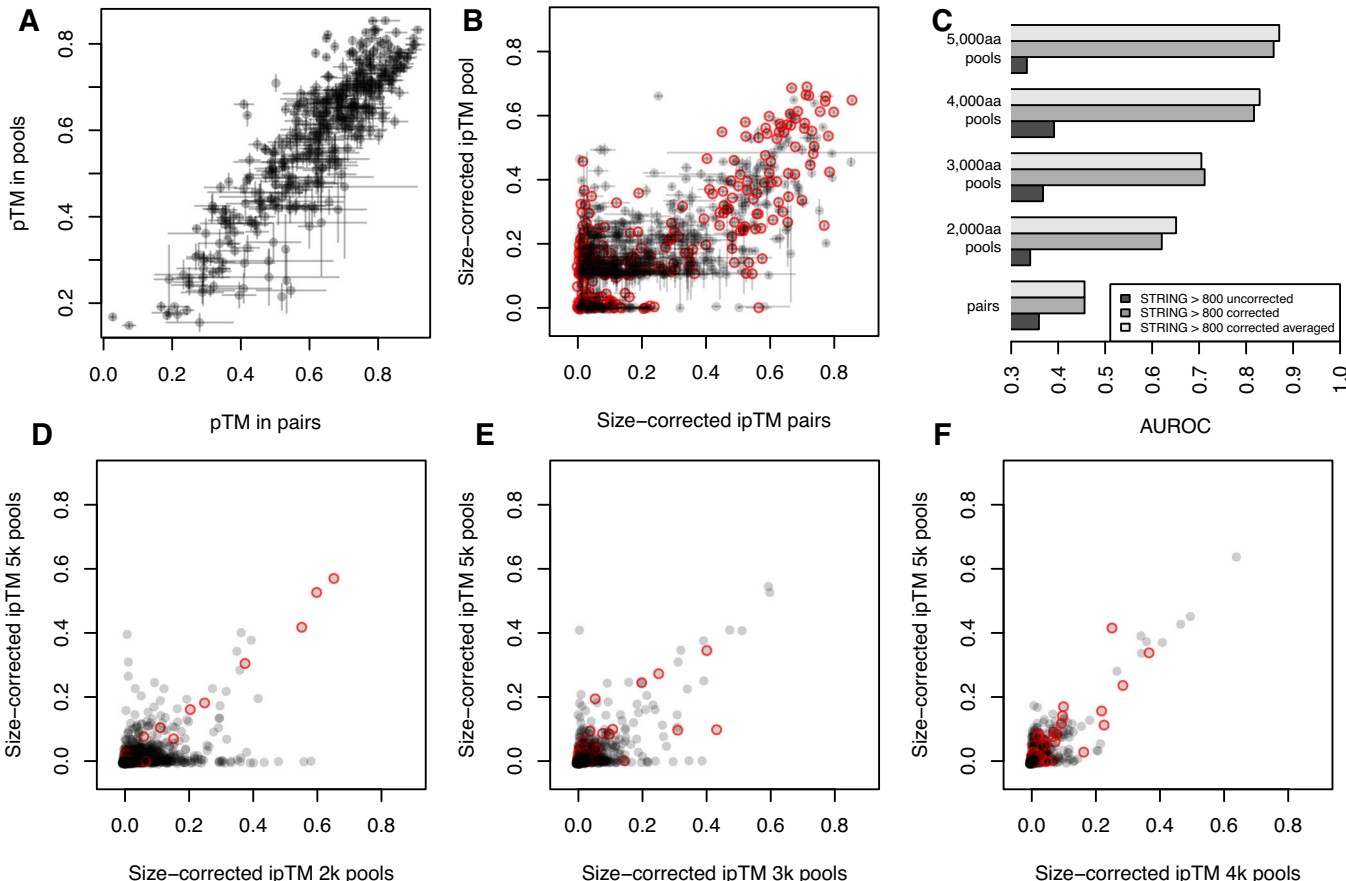

**Figure EV7.  Larger pool sizes are more predictive of known PPI and exhibit fewer false-positive hits.**

Red circles in (**B**, **D–F**) represent protein pairs with STRING experimental scores >800. (**A**) Average pTM scores for all 418 proteins for which we had both pooled and paired data. pTMs were highly correlated between the pairs and pools (Pearson's $r = 0.882$) and did not exhibit systematic differences (median difference = 0.039), indicating that folding in pools does not affect AlphaFold3's ability to predict monomer structures. (**B**) Size-corrected ipTMs of protein pairs with high scores in the pools are well correlated. (Pearson's $r = 0.725$, 942 pairs), though ~0 to 0.2 lower in pools. (**C**) AUROC for ~4500 protein pairs assayed using randomly generated pools of different sizes. Raw, corrected, and corrected averaged data is shown. (**C**) ROC curves for the different pool sizes. (**D–F**) Comprehensive pooled ipTMs compared to ipTMs from 2000 aa (**D**), 3000 aa (**E**) and 4000 aa (**F**) pools.

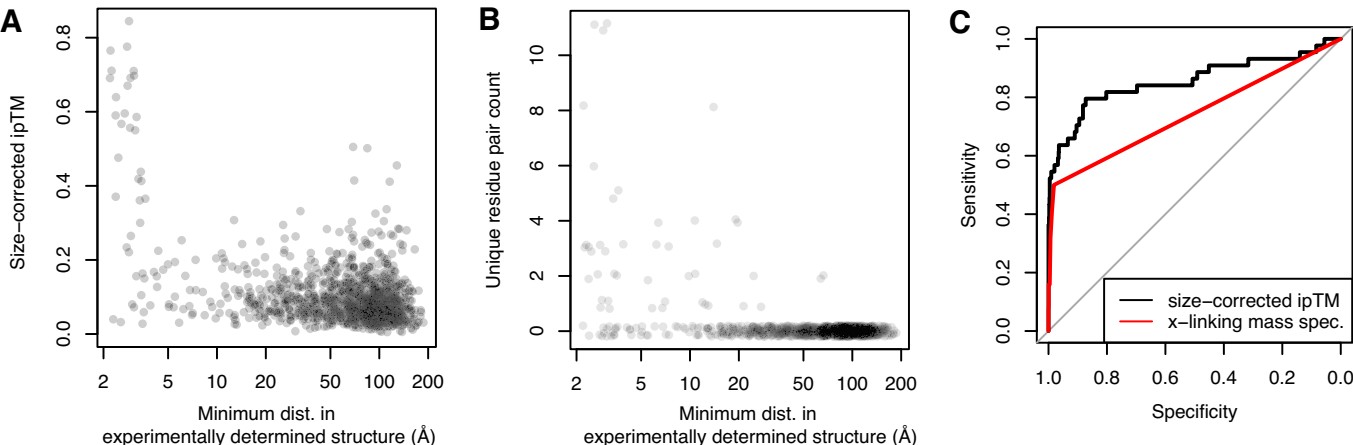

**Figure EV8. Additional information about the ribosome structure prediction.**

(A) Plot of size-corrected ipTM versus minimum distance for all ribosomal protein pairs. (B) Plot of number of unique cross-links versus minimum distance for all ribosomal protein pairs. Some of the additional cross-links in the XL-MS data may be due to the linker length of the cross-linking reagents. (C) AUROC curve showing the performance of XL-MS (red) and size-corrected ipTM for identifying ribosomal protein pairs within 5 Å.

