## [Peer Review File · Molecular Systems Biology]

Predicting the protein interaction landscape of a free-living bacterium with pooled-AlphaFold3

Horia Todor, Lili Kim, Jurgen Janes, Hannah Burkhart, Seth Darst, Pedro Beltrao, and Carol Gross

Corresponding author(s): Horia Todor (horia.todor@ucsf.edu)

Review Timeline:

Submission Date:	24th Sep 25
Editorial Decision:	3rd Nov 25
Revision Received:	5th Dec 25
Editorial Decision:	16th Dec 25
Revision Received:	17th Dec 25
Accepted:	8th Jan 26

Editor: Jingyi Hou

Transaction Report:

3rd Nov 2025

Manuscript Number: MSB-2025-13376

Title: Predicting the protein interaction landscape of a free-living bacterium with pooled-AlphaFold3

Author: Horia Todor

Lili Kim

Jurgen Janes

Hannah Burkhart

Seth Darst

Pedro Beltrao

Carol Gross

Dear Horia,

Thank you for submitting your work to Molecular Systems Biology. We have now heard back from the three reviewers who agreed to evaluate your manuscript. As you will see from the comments below that they find the method novel and interesting. They raise, however, several important points, which should be convincingly addressed in a revision of this work.

I think that the recommendations of the reviewers are rather straightforward so there is no need to repeat the points listed below. All issues raised by the reviewers need to be satisfactorily addressed.

As you may already know, our editorial policy allows in principle a single round of major revision so it is essential to provide responses to the reviewers' comments that are as complete as possible. Please feel free to contact me in case you would like to discuss in further detail any of the issues raised by the reviewers.

On a more editorial level, we would ask you to address the following issues:

- Please provide a .docx formatted version of the manuscript text (including legends for main figures, EV figures and tables). Please make sure that the changes are highlighted to be clearly visible.
- Please provide individual production quality figure files as .eps, .tif, .jpg (one file per figure).
- Please provide a .docx formatted letter INCLUDING the reviewers' reports and your detailed point-by-point responses to their comments. As part of the EMBO Press transparent editorial process, the point-by-point response is part of the Review Process File (RPF), which will be published alongside your paper.
- Please note that all corresponding authors are required to supply an ORCID ID for their name upon submission of a revised manuscript.
- We replaced Supplementary Information with Expanded View (EV) Figures and Tables that are collapsible/expandable online (see examples in <http://msb.embopress.org/content/11/6/812>). A maximum of 5 EV Figures can be typeset. EV Figures should be cited as 'Figure EV1, Figure EV2' etc... in the text and their respective legends should be included in the main text after the legends of regular figures.

Additional Tables/Datasets should be labeled and referred to as Table EV1, Dataset EV1, etc. Legends have to be provided in a separate tab in case of .xls files. Alternatively, the legend can be supplied as a separate text file (README) and zipped together with the Table/Dataset file.

For the figures and tables that you do NOT wish to display as Expanded View figures, they should be bundled together with their legends in a single PDF file called *Appendix*, which should start with a short Table of Content. Each legend should be below the corresponding Figure/Table in the Appendix. Appendix figures and tables should be referred to in the main text as: "Appendix Figure S1, Appendix Figure S2, Appendix Table S1" etc. See detailed instructions regarding expanded view here: <https://www.embopress.org/page/journal/17444292/authorguide#expandedview>.

- Before submitting your revision, primary datasets (and computer code, where appropriate) produced in this study need to be deposited in an appropriate public database (see <http://msb.embopress.org/authorguide> - dataavailability <https://www.embopress.org/page/journal/17444292/authorguide#dataavailability>). Please remember to provide a reviewer password if the datasets are not yet public. The accession numbers and database should be listed in a formal "Data Availability" section (placed after Materials & Method) that follows the model below (see also <https://www.embopress.org/page/journal/17444292/authorguide#dataavailability>). Please note that the Data Availability Section is restricted to new primary data that are part of this study.

Data availability

- RNA-Seq data: Gene Expression Omnibus GSE46843 (<https://www.ncbi.nlm.nih.gov/geo/query/acc.cgi?acc=GSE46843>)

- [data type]: [name of the resource] [accession number/identifier/doi] ([URL or identifiers.org/DATABASE:ACCESSION])

Additional information on source data and instruction on how to label the files are available

- Our journal encourages inclusion of *data citations in the reference list* to directly cite datasets that were re-used and obtained from public databases. Data citations in the article text are distinct from normal bibliographical citations and should directly link to the database records from which the data can be accessed. In the main text, data citations are formatted as follows: "Data ref: Smith et al, 2001". In the Reference list, data citations must be labeled with "[DATASET]". A data reference must provide the database name, accession number/identifiers and a resolvable link to the landing page from which the data can be accessed at the end of the reference. Further instructions are available at .

- We updated our journal's competing interests policy in January 2022 and request authors to consider both actual and perceived competing interests. Please review the policy <https://www.embopress.org/competing-interests> and update your competing interests if necessary.

Please use the heading "Disclosure statement and competing interests".

- All Materials and Methods need to be described in the main text using our 'Structured Methods' format. According to this format, the Methods section includes a Reagents and Tools Table (listing key reagents, experimental models, software and relevant equipment and including their sources and relevant identifiers) followed by a Methods and Protocols section describing the methods, ideally using a step-by-step protocol format. The aim is to facilitate adoption of the methodologies across labs. Please download and fill our Reagents and Tools Table template (.docx), which you can find in our author guidelines:

<https://www.embopress.org/page/journal/17444292/authorguide#structuredmethods>.

-Regarding data quantification:

Please ensure to specify the name of the statistical test used to generate error bars and P values, the number (n) of independent experiments (please specify technical or biological replicates) underlying each data point and the test used to calculate p-values in each figure legend. Discussion of statistical methodology can be reported in the materials and methods section, but figure legends should contain a basic description of n, P and the test applied.

Graphs must include a description of the bars and the error bars (s.d., s.e.m.).

- Please provide a "standfirst text" summarizing the study in one or two sentences (approximately 250 characters, including space), three to four "bullet points" highlighting the main findings and a "synopsis image" (550px width and 400-600 px height, PNG format) to highlight the paper on our homepage.

Here are a couple of examples:

<https://www.embopress.org/doi/10.15252/msb.20199356>

<https://www.embopress.org/doi/10.15252/msb.20209475>

<https://www.embopress.org/doi/10.15252/msb.209495>

When you resubmit your manuscript, please download our CHECKLIST (<https://www.embopress.org/pb-assets/embosite/EMBO%20Press%20Author%20Checklist-1642513524327.xlsx>) and include the completed form in your submission.

Please note that the Author Checklist will be published alongside the paper as part of the transparent process (<https://www.embopress.org/page/journal/17444292/authorguide#transparentprocess>).

If you feel you can satisfactorily deal with these points and those listed by the referees, you may wish to submit a revised version of your manuscript. Please attach a covering letter giving details of the way in which you have handled each of the points raised by the referees. A revised manuscript will be once again subject to review and you probably understand that we can give you no guarantee at this stage that the eventual outcome will be favorable.

I look forward to receiving your revised manuscript soon.

Sincerely,

Jingyi

Jingyi Hou, PhD
Senior Editor
Molecular Systems Biology

We realize that it is difficult to revise to a specific deadline. In the interest of protecting the conceptual advance provided by the work, we recommend a revision within 3 months (1st Feb 2026). Please discuss the revision progress ahead of this time with the editor if you require more time to complete the revisions. Use the link below to submit your revision:

*** PLEASE NOTE *** As part of the EMBO Press transparent editorial process initiative (see our Editorial at <https://dx.doi.org/10.1038/msb.2010.72>), Molecular Systems Biology publishes online a Review Process File with each accepted manuscripts. This file will be published in conjunction with your paper and will include the anonymous referee reports, your point-by-point response and all pertinent correspondence relating to the manuscript. If you do NOT want this File to be published, please inform the editorial office at contact@molsystbiol.org within 14 days upon receipt of the present letter.

Reviewer #1:

In this paper, the authors find that AlphaFold3's ability to identify interacting proteins can be improved if:

1. The ipTM score is normalised by length
2. Potential binary interactions are tested in Pools of several proteins.

Both these results are novel and interesting. Therefore this paper is suitable for publication

Comments, suggestions

1. It would be interesting to also test this with AlphaFold2.3 - to see if the same scaling and pooling behaviour is observed. In particular, the PAEs are calculated differently (as a separate head rather than unrolling the diffusion model), so it is not clear whether that is the case. A small-scale test would suffice.
2. What happens for larger complexes, if you run a complex with X chains and then add some non interacting ones, does the intra-chain ipTMs (also reported by AF3) improve? How does length scaling happens?

Reviewer #2:

Summary

This paper describes a novel computational methodology for predicting protein-protein interactions (PPIs) utilizing AlphaFold3. The authors use a pooled co-folding approach (multiple proteins are added to the same prediction job) to determine whether two proteins form a PPI, rather than the traditional pairwise approach (only the two proteins of interest are co-folded). They applied this methodology to the *Mycoplasma genitalium* proteome, generating 2,027 pooled AlphaFold3 jobs to test all 113,050 unique protein pairs, demonstrating a considerable reduction in computational cost. The authors also identified a size-dependent bias in interface scores (ipTM values) in AlphaFold3 and developed a correction framework. Using these size-corrected interface-predicted TM-scores and correlation profiles, the model achieved an AUROC of up to 0.92 when benchmarked against the STRING database. Notably, this method also uncovered previously uncharacterized interactions in *M. genitalium*, including a heterodimeric RNase Y complex, a truncated trigger factor potentially involved in secretion, and a kinase-cell division interaction between EzrA and PrkC.

General remarks

Overall, the study is convincing and well supported by benchmarking against the STRING database and structural data. The pooled AlphaFold3 approach clearly improves scalability and reduces false positives compared to pairwise predictions. While some conclusions, such as the accuracy of predicted novel interactions, would benefit from experimental validation, the

computational evidence is strong. This work represents both a technical and conceptual advance, making proteome-wide PPI prediction feasible on standard resources and revealing a critical size-dependent bias in AlphaFold3 scoring.

Major points

It is not clear why the authors rely exclusively on the corrected ipTM as the primary interaction metric. Since AlphaFold3 outputs various interface-related measures, such as ipTM, PAE, etc., it would be helpful to briefly explain why ipTM was selected. Could combining multiple metrics improve predictive performance?

The description of how the pooled proteome-wide runs were generated could be clarified. Providing details in the Methods section and code on how pairwise matching of all unique protein pairs was ensured while minimizing the total job count would allow readers to easily apply this novel approach.

The text mentions that 2,027 pooled AlphaFold3 jobs were run, but it is not specified whether all runs completed successfully. We tested the pooled run methodology on *Mycobacterium tuberculosis* PPIs, however, some of the runs failed. Do the authors have insight into why this might have occurred? Clarifying whether any of the 2,027 jobs failed or required reruns would greatly support reproducibility in other systems.

The description of the size-dependent correction is somewhat challenging to follow. While the relevant details appear across the main text, Supplementary Text 1, and Figure S3, they are scattered and not fully summarized in a single place. Including a concise explanation in the Methods section and sharing the code would make the correction more straightforward to understand and reproduce.

While the corrected ipTM scores correlate well with STRING-derived PPIs, it would be interesting to know whether these corrected scores also align with known binding affinities or experimentally validated interaction data. A brief comparison or citation further supports the biological relevance of the correction.

Minor points

Several terms, such as "pool," "batch," and "replicate," appear to be used interchangeably. Defining these terms clearly at the outset would help readers follow the text more easily.

Consider specifying AlphaFold3 default parameters for transparency (e.g., number of models, relaxation settings).

Figure 1A: Excellent and intuitive illustration of the pooled AlphaFold3 workflow.

Figure S7B: The y-axis label is unclear; please specify whether it refers to system size or another variable.

Reviewer #3:

Todor et al. present an all vs all set of AlphaFold3 predictions for the entire pairwise interactome of *Mycoplasma genitalium*. They develop a powerful strategy of generating AlphaFold models in pools of up to 5k tokens (i.e. amino acids). This reduces the total number of jobs, reduces overall runtime, and shows improvements in the false positive rate of high throughput AF predictions for protein interactions. Further, they identify a size bias in the ipTM score generated by AF3 which they correct for and improve scoring. This work overall is a valuable contribution to the field. I have only a few comments that should be addressed prior to publication.

Major:

- 1) The improvement of pooled vs pairwise is quite striking. It is unclear to me what the mechanism is for improvement. The authors state there is a benefit from "competition". Does this mean general competition where a pool of random proteins will improve the modeling or are there specific proteins in the pool that are increasing the competition for a binding site? It would seem the authors have the dataset to test some of these hypotheses given they have tested thousands of protein pairs in separate pools and identified variability between the pools.
- 2) The authors test whether the variability between pools is due to false-positives or false negatives. It is unclear, as written, of the logic of why using minimum vs maximum ipTM values tests this. The authors should give better intuition for this specific section.
 - a. Related, it seems the analysis in Fig2E shows minimal confident corrected ipTM values for the pool vs the pairs. This suggests pooling reduces false positives, at least with respect to pairwise model computing. Can the authors comment?
- 3) There are several other groups that have identified pathologies in the ipTM as the authors mention in the discussion. Do the corrections suggested by others deal with this size bias (explicitly or implicitly)? Can these other methods be improved by correcting for size? Further, how much of the size bias is due to disorder content as partly suggested by others (Dunbrack 2025)?
- 4) The methods section is extremely limited.
 - a. For example, it is unclear how proteins were pooled to minimize the total number of pools. A full description is necessary if someone wants to replicate or to apply this to another organism. A link to code would be beneficial.
 - b. It is also not clear how the "representative set" of ~4,500 pairwise interactions was created for figure 2E. The text points to the

methods section but there is no description there.

Minor:

- 1) The legend of Fig1A has a cryptic job id, "250310_mgen_allbyall_627". What does this refer to?
- 2) On page 12, last paragraph, the nomenclature "job allbyall" was not introduced and needs clarification.
- 3) Figure 5 should have the corrected ipTMs listed in the figure for each model described.
- 4) TableS3_representative_pairs and high_ipTm_pairs_rerun do not have column headers. TableS2 also needs more descriptive column headers.

Summary

We would like to thank all three reviewers for their insightful and positive reviews of our manuscript. Motivated by their suggestions, we rewrote certain sections of the text, fixed numerous minor ambiguities, and added additional details to the methods (including a link to the pooling and analysis code), which we feel greatly strengthened and clarified our manuscript.

Changes in the revised manuscript are in red font, and detailed responses to all reviewer comments are below.

Reviewer #1:

In this paper, the authors find that AlphaFold3's ability to identify interacting proteins can be improved if:

- 1. The ipTM score is normalised by length**
- 2. Potential binary interactions are tested in Pools of several proteins.**

Both these results are novel and interesting. Therefore this paper is suitable for publication

We thank the reviewer for their succinct and positive feedback.

Comments, suggestions

1. It would be interesting to also test this with AlphaFold2.3 - to see if the same scaling and pooling behaviour is observed. In particular, the PAEs are calculated differently (as a separate head rather than unrolling the diffusion model), so it is not clear whether that is the case. A small-scale test would suffice.

This is a very interesting suggestion. We interpret this suggestion in two parts. First, is the ipTM-length relationship shared by other structure prediction programs that use other methods for determining PAEs? Second, is the improvement in PPI prediction shared by other structure prediction programs?

- Is the ipTM-length relationship shared by other structure prediction programs? The length-ipTM relationship appears to be shared by AF2 and AF3. This is shown in Fig. S3 where we plot ipTM by length for several datasets in the literature. To make this clearer, the revised figure legend now contains specific AlphaFold2 version numbers for each dataset.
- Is the improvement in PPI prediction shared by other structure prediction programs? Unfortunately, AlphaFold2.3 is not suitable for pooled-PPI because of its performance characteristics: publicly available performance benchmarks suggest AlphaFold2.3 is at least ~5-10x slower than AlphaFold3 for large (~5,000aa) jobs, implying a runtime of 5-10hrs per pool (!) on a Nvidia A100 80gb GPU.

We added a sentence to the introduction explaining our rationale for choosing AF3:

“Currently, AlphaFold3 is singularly well-suited for this approach because its computational efficiency enables high-throughput folding of large jobs (e.g., 5,000aa pools) on a single 80gb GPU, though this may change rapidly (e.g., Litfin et al, 2025).”

2. What happens for larger complexes, if you run a complex with X chains and then add some non-interacting ones, does the intra-chain ipTMs (also reported by AF3) improve? How does length scaling happens?

We thank the reviewer for this suggestion. To assess how running in pools affects the pTM (i.e., intra-chain ipTM), we compared the pTMs for all 418 proteins for which we had both pooled and paired data. pTMs were highly correlated between the pairs and pools ($r = 0.88$) and did not exhibit systematic differences (median difference = 0.04), indicating that folding in pools does not affect AlphaFold3’s ability to predict monomer structures. We have added this data as Fig. S7A and reference it in the text (changes underlined).

“Consistent with this, size-corrected ipTMs of protein pairs were poorly correlated between the paired and pooled approaches (Pearson’s $r = 0.157$), despite similar per-chain pTMs (Figure S7A).”

For protein pairs with high ipTM scores in pools, the ipTMs from pools are generally similar to those of individually folded protein but 0 to 0.2 smaller than ipTMs for pairs (Fig. S7B). The relationship between length and ipTM is conserved for pairs and pools (see the updated Fig. S3).

Reviewer #2:

Summary

This paper describes a novel computational methodology for predicting protein-protein interactions (PPIs) utilizing AlphaFold3. The authors use a pooled co-folding approach (multiple proteins are added to the same prediction job) to determine whether two proteins form a PPI, rather than the traditional pairwise approach (only the two proteins of interest are co-folded). They applied this methodology to the Mycoplasma genitalium proteome, generating 2,027 pooled AlphaFold3 jobs to test all 113,050 unique protein pairs, demonstrating a considerable reduction in computational cost.

The authors also identified a size-dependent bias in interface scores (ipTM values) in AlphaFold3 and developed a correction framework. Using these size-corrected interface-predicted TM-scores and correlation profiles, the model achieved an AUROC of up to 0.92 when benchmarked against the STRING database. Notably, this method also uncovered previously uncharacterized interactions in M. genitalium, including a

heterodimeric RNase Y complex, a truncated trigger factor potentially involved in secretion, and a kinase-cell division interaction between EzrA and PrkC.

General remarks

Overall, the study is convincing and well supported by benchmarking against the STRING database and structural data. The pooled AlphaFold3 approach clearly improves scalability and reduces false positives compared to pairwise predictions. While some conclusions, such as the accuracy of predicted novel interactions, would benefit from experimental validation, the computational evidence is strong. This work represents both a technical and conceptual advance, making proteome-wide PPI prediction feasible on standard resources and revealing a critical size-dependent bias in AlphaFold3 scoring.

We thank the reviewer for their positive feedback.

Major points

It is not clear why the authors rely exclusively on the corrected ipTM as the primary interaction metric. Since AlphaFold3 outputs various interface-related measures, such as ipTM, PAE, etc., it would be helpful to briefly explain why ipTM was selected.

ipTM is the most commonly used metric for evaluating protein complexes: its prominence in the AlphaFold3 interface makes it the “front-line number”. Because of this, it is the metric that most scientists, especially those outside the protein structure prediction field, are most familiar with. We have added a sentence describing our choice to focus on ipTM where it is first introduced (changes underlined):

“AlphaFold3 evaluates the accuracy of predicted complexes using several metrics, the most common of which is the interface predicted template modeling (ipTM) score.”

Could combining multiple metrics improve predictive performance?

We think it is likely that combining multiple metrics may improve AUROC. We prefer not to chase these optimizations for two reasons: First, it is unclear how broadly applicable such optimizations may be outside of the *M. genitalium* proteome and/or the STRING database of known positive interactions. Second, we worry that such optimizations will obfuscate the two central messages of the manuscript - that pooling drastically decreases false positives and that ipTMs are biased by the size of the interacting proteins.

The description of how the pooled proteome-wide runs were generated could be clarified. Providing details in the Methods section and code on how pairwise matching of all unique protein pairs was ensured while minimizing the total job count would allow readers to easily apply this novel approach.

We have added additional information on how the pools were generated in the methods section “Preparing pools” as well as a github link to our code. The relevant section now includes the following sentences (changes underlined):

“The genome of *M. genitalium* was downloaded from NCBI (GenBank: L43967.2, Table EV1). Proteins were pooled to minimize the number of pools required to co-fold all protein pairs at least once while keeping all jobs <5,000 tokens. This was accomplished using a greedy algorithm available at <https://github.com/horiatodor/pooled-af3>. Briefly, pools are initially completely random, but become increasingly biased for including proteins that have the largest number of missing interactions.”

The text mentions that 2,027 pooled AlphaFold3 jobs were run, but it is not specified whether all runs completed successfully. We tested the pooled run methodology on *Mycobacterium tuberculosis* PPIs, however, some of the runs failed. Do the authors have insight into why this might have occurred? Clarifying whether any of the 2,027 jobs failed or required reruns would greatly support reproducibility in other systems.

We clarified that all 2,027 jobs ran successfully in the methods.

Anecdotally, large jobs on alphafoldserver.com fail (with a “model inference failed” error) when server load is high. The solution is simply to resubmit failed jobs until they successfully run. Since this phenomenon involves the setup of the private AlphaFoldServer (and may change without warning), we consider it outside the scope of our manuscript.

The description of the size-dependent correction is somewhat challenging to follow. While the relevant details appear across the main text, Supplementary Text 1, and Figure S3, they are scattered and not fully summarized in a single place. Including a concise explanation in the Methods section and sharing the code would make the correction more straightforward to understand and reproduce.

We agree with this suggestion, and have now added a concise description of the size correction in its own section “Size correction of ipTM scores” in the methods as well as a link to the code. The section reads (changes underlined):

“To determine the effect of summed protein size on ipTM, we performed a robust linear regression using the `robustbase::lmrob` function, which computes a MM-type regression estimator (a robust and efficient estimator with a ~50% breakdown point and 95% efficiency). A robust estimator was used to mitigate the influence of true-positive interactions on the regression. The linear regression was performed on the square-root of the summed size of the two proteins (in amino acids) to calculate an “expected ipTM” for each protein pair. For the *M. genitalium* dataset, $\text{expected_ipTM} = -0.036255571 + 0.004470512 * \sqrt{\text{aa_in_protein1} + \text{aa_in_protein2}}$. We expect these coefficients to be similar for other datasets. Expected ipTM”

was subtracted from the observed ipTM to generate the “size-corrected ipTM”. Code for implementing this correction is available at <https://github.com/horiatodor/pooled-af3>.”

While the corrected ipTM scores correlate well with STRING-derived PPIs, it would be interesting to know whether these corrected scores also align with known binding affinities or experimentally validated interaction data. A brief comparison or citation further supports the biological relevance of the correction.

We are not aware of any genome-scale binding affinity data in this organism. Moreover, AlphaFold3 itself was trained only on the structure of protein complexes, and not on binding affinity. Thus, it seems unlikely that ipTMs (corrected or not) would be related to binding affinity. Therefore, we feel that a thorough investigation of this issue is outside the scope of our manuscript.

Minor points

Several terms, such as "pool," "batch," and "replicate," appear to be used interchangeably. Defining these terms clearly at the outset would help readers follow the text more easily.

We thank the reviewer for pointing out the ambiguity around our use of “job”, “replicate”, and “pool” (the word “batch” does not appear in the text).

We have gone through the text and made a number of edits to clarify our meaning, including removing all references to “replicate”, and defining “job” and “pool”. Briefly, a job is a single AlphaFold3 run and can contain a single protein, a pair of proteins, or a pool of proteins.

Consider specifying AlphaFold3 default parameters for transparency (e.g., number of models, relaxation settings).

We have added the AlphaFold3 default parameters to the methods section. They are: 1 seed; 5 diffusion samples, 10 recycles and template cutoff date of September 30, 2021.

Figure 1A: Excellent and intuitive illustration of the pooled AlphaFold3 workflow.

We thank the reviewer for this positive comment.

Figure S7B: The y-axis label is unclear; please specify whether it refers to system size or another variable.

We have clarified the axis labels for this figure (now Figure EV7C).

Reviewer #3:

Todor et al. present an all vs all set of AlphaFold3 predictions for the entire pairwise interactome of *Mycoplasma genitalium*. They develop a powerful strategy of generating AlphaFold models in pools of up to 5k tokens (i.e. amino acids). This reduces the total number of jobs, reduces overall runtime, and shows improvements in the false positive rate of high throughput AF predictions for protein interactions. Further, they identify a size bias in the ipTM score generated by AF3 which they correct for and improve scoring. This work overall is a valuable contribution to the field. I have only a few comments that should be addressed prior to publication.

We thank the reviewer for their positive feedback and helpful suggestions.

Major:

1) The improvement of pooled vs pairwise is quite striking. It is unclear to me what the mechanism is for improvement. The authors state there is a benefit from "competition". Does this mean general competition where a pool of random proteins will improve the modeling or are there specific proteins in the pool that are increasing the competition for a binding site? It would seem the authors have the dataset to test some of these hypotheses given they have tested thousands of protein pairs in separate pools and identified variability between the pools.

We agree with the reviewer that this is an interesting question, but the "why" of neural networks is a notoriously difficult problem. Moreover, using the variability between pools to address this question is problematic because of intrinsic variability in the output of AlphaFold3 when running even the same job with different random seeds (see Fig. 2F). Nonetheless, this is an important question. Briefly, the improvement is likely primarily due to:

A. General competition effect. Consider a protein with a "sticky" pocket that is incorrectly predicted to bind any protein (100% false positive rate). If only 1 protein can fit in that pocket at any given time, pooling 10 proteins would decrease the observed false positive rate from 100% to 10%.

B. Spatial constraints. The geometry of a pooled complex makes it impossible for all protein pairs to interact, forcing AlphaFold3 to prioritize some interactions over others.

We have revised the last sentence of the introduction to include this speculation (changes underlined):

"Pooled-PPI prediction decreases overall run-time (~2-fold, depending on hardware), the number of individual jobs (up to 300-fold, depending on pool size), and increases the accuracy of PPI predictions through general competition and spatial constraints, as previously demonstrated in small scale peptide-target screens (Chang & Perez, 2023; Vosbein *et al*, 2024; Mondal *et al*, 2024)."

2) The authors test whether the variability between pools is due to false-positives or false negatives. It is unclear, as written, of the logic of why using minimum vs maximum ipTM values tests this. The authors should give better intuition for this specific section.

We have added the following two sentences to the section to better explain our logic for this specific section.

“We reasoned that if discrepant values were solely due to false positives, taking the minimum value should improve predictive performance. Similarly, if discrepant values were due solely to false negatives, taking the maximum value should improve predictive performance.”

a. Related, it seems the analysis in Fig2E shows minimal confident corrected ipTM values for the pool vs the pairs. This suggests pooling reduces false positives, at least with respect to pairwise model computing. Can the authors comment?

All AlphaFold3 ipTM values shown in the manuscript are the average of all 5 diffusion samples for that protein pair. We have updated the figure legend to reflect this. We agree with the reviewer that pooling reduces false positives, and the manuscript already comments on this extensively in the section that describes Fig. 2E.

“Together, our comparison between pooled-AlphaFold3 and individually folded pairs suggests a low false-positive rate for the pooled approach vis-a-vis individually co-folded protein pairs.”

“The increased AUROC was driven primarily by a decrease in false-positive hits (Fig. S7D-F).”.

3) There are several other groups that have identified pathologies in the ipTM as the authors mention in the discussion. Do the corrections suggested by others deal with this size bias (explicitly or implicitly)?

This is an important point and we acknowledge these attempts in the text, while noting that they do not explicitly deal with or identify the size bias in ipTM scores:

“Although several shortcomings of ipTM scores have been previously described (Kim *et al*, 2024; Dunbrack, 2025; Varga *et al*, 2025), the size dependence of non-interacting ipTM scores has not been explicitly noted.”

Moreover, our impression is that these corrections overwhelmingly deal with false-negatives (where the ipTM is low, but the interface is correct), not false positives (where the ipTM is high, but the interaction is not present).

Can these other methods be improved by correcting for size?

This is an interesting question but outside the scope of our manuscript, particularly since Dunbrack 2025 and Varga et al 2025 focus on ipTM as a measure of structural accuracy not PPI determination.

Further, how much of the size bias is due to disorder content as partly suggested by others (Dunbrack 2025)?

Most proteins have a low disordered fraction and there is very little correlation between fraction disordered and ipTM for the 4,560 random pairs we ran (first panel). Compare to size-ipTM (second panel).

4) The methods section is extremely limited.

a. For example, it is unclear how proteins were pooled to minimize the total number of pools. A full description is necessary if someone wants to replicate or to apply this to another organism. A link to code would be beneficial.

We have added a description of the pooling strategy as well as a link to the code we used to the manuscript.

b. It is also not clear how the "representative set" of ~4,500 pairwise interactions was created for figure 2E. The text points to the methods section but there is no description there.

We have added a description of how the representative set of 4,500 pairwise interactions selected. The sentence now reads:

“To compare pairwise predictions to our pooled approach, we randomly selected 96 proteins (~20% of the *M. genitalium* proteome), used AlphaFold3 to individually co-fold all possible pairs of these proteins (4,560 pairs, Methods, Table S3), and ascertained their ability to predict known PPIs.”

Minor:

We thank the reviewer for all the comments below, which pointed out needed clarifications.

1) The legend of Fig1A has a cryptic job id, "250310_mgen_allbyall_627". What does this refer to?

"250310_mgen_allbyall_627" is one of the pooled AlphaFold3 jobs run for this study. The contents of all pooled jobs are listed in Table S2. We added a reference to Table S2 and extra context to the Figure 1 legend.

2) On page 12, last paragraph, the nomenclature "job allbyall" was not introduced and needs clarification.

We updated the job title in the text to be consistent with those in Fig. 1A and Table S2 and added language to introduce the title.

3) Figure 5 should have the corrected ipTMs listed in the figure for each model described.

All size-corrected ipTMs are now displayed in the appropriate figure.

4) TableS3_representative_pairs and high_ipTM_pairs_rerun do not have column headers. TableS2 also needs more descriptive column headers.

We thank the reviewer for noticing these omissions. We added column headers to Table S3 and S2 where needed and expanded or added keys for important information about sheets and column headers in Table S3 and Table S2.

16th Dec 2025

Manuscript Number: MSB-2025-13376R

Title: Predicting the protein interaction landscape of a free-living bacterium with pooled-AlphaFold3

Author: Horia Todor

Lili Kim

Jurgen Janes

Hannah Burkhart

Seth Darst

Pedro Beltrao

Carol Gross

Dear Horia,

Thank you for sending us your revised manuscript. We have now heard back from the two reviewers who were asked to re-evaluate your study. As you will see below, the reviewers are both satisfied with the modifications made. Before we can formally accept your manuscript, we would ask you to address the following editorial-level issues:

1. Remove "Authors' contribution" section from the manuscript file.
2. The Data Availability statement needs to be placed before "Acknowledgements".
3. The section "EXPANDED VIEW MATERIALS" should be removed from the manuscript file.
4. Please provide an institutional email address for the corresponding author rather than a personal email address.
5. Funding information should be included in the Acknowledgments section. A separate Funding heading should not be used.
6. The current "SUPPLEMENTARY TEXT 1" part should be removed from the manuscript file and uploaded as a separate PDF file labeled "Appendix". The Appendix file should begin with a title page ("Appendix for [manuscript title]"), followed by a Table of Contents that includes page numbers for all listed items. Please use the nomenclature Appendix Figure S1 and update all corresponding callouts in the manuscript accordingly.
7. Eight EV tables have been uploaded (source file names labeled Table S1, etc.). Some tables contain incorrect legends in the titles. For example, Table EV6 displays "Table S5" when opened, and Table EV8 displays "Table S7" in the legend. Please review and correct these inconsistencies.

All tables except Table EV6 are complex and should be updated to Dataset EV1-EV7 across all instances, including source file names, system titles, table legends, and manuscript callouts.

References are not permitted in Table EV6; please remove all references from this table. After the other EV tables have been renamed as EV datasets, Table EV6 should be renumbered as Table EV1, and the callouts need to be updated accordingly.

8. Please provide a "standfirst text" summarizing the study in one or two sentences (approximately 250 characters, including space), three to four "bullet points" highlighting the main findings and a "synopsis image" (550px width and 400-600 px height, PNG format) to highlight the paper on our homepage. Please refer to published papers for examples.

9. Please use our Reagents and Tools Table template (.docx), which you can find in our author guidelines: <https://link.springer.com/journal/44320/submission-guidelines#structuredmethods>.

10. Please address the following issues in figure legends:

- Please note that the box plots need to be defined in terms of minima, maxima, centre, bounds of box and whiskers, and percentile in the legends of figures 1D, EV1 B, C
- Please note that information related to n is missing in the legends of figures 2C, E, F; EV1 C, EV4 B, C.
- Please note that the error bars are not defined in the legends of figures 2C, E, F; EV4 B, C

Thank you for submitting this interesting paper to Molecular Systems Biology.

Kind regards,
Jingyi

Jingyi Hou, PhD
Senior Editor
Molecular Systems Biology

*** PLEASE NOTE *** As part of the EMBO Press transparent editorial process initiative (see our Editorial at <https://dx.doi.org/10.1038/msb.2010.72> , Molecular Systems Biology will publish online a Review Process File to accompany accepted manuscripts. When preparing your letter of response, please be aware that in the event of acceptance, your cover letter/point-by-point document will be included as part of this File, which will be available to the scientific community. More information about this initiative is available in our Instructions to Authors. If you have any questions about this initiative, please contact the editorial office (msb@embo.org).

Reviewer #2:

The authors have clearly and comprehensively addressed feedback. Our concerns have been addressed. This is a strong and important manuscript that deserves publication.

Reviewer #3:

The authors have addressed my concerns and I believe this work will be an extremely valuable contribution to the field.

Title: Predicting the protein interaction landscape of a free-living bacterium with pooled-AlphaFold3**Response to editorial and reviewer comments**

Author responses are marked in red.

Editorial Comments

Remove "Authors' contribution" section from the manuscript file.

The Author Contribution section has been removed.

The Data Availability statement needs to be placed before "Acknowledgements".

The Data Availability statement has been placed before Acknowledgements.

The section "EXPANDED VIEW MATERIALS" should be removed from the manuscript file.

The Expanded View Materials section has been removed.

Funding information should be included in the Acknowledgments section. A separate Funding heading should not be used.

Funding information has been incorporated into the Acknowledgements.

The current "SUPPLEMENTARY TEXT 1" part should be removed from the manuscript file and uploaded as a separate PDF file labeled "Appendix". The Appendix file should begin with a title page ("Appendix for [manuscript title]"), followed by a Table of Contents that includes page numbers for all listed items. Please use the nomenclature Appendix Figure S1 and update all corresponding callouts in the manuscript accordingly.

Supplementary Text 1 has been removed from the manuscript and uploaded as a separate Appendix PDF file with appropriate formatting. Callouts have been updated.

Eight EV tables have been uploaded (source file names labeled Table S1, etc.). Some tables contain incorrect legends in the titles. For example, Table EV6 displays "Table S5" when opened, and Table EV8 displays "Table S7" in the legend. Please review and correct these inconsistencies.

All tables except Table EV6 are complex and should be updated to Dataset EV1-EV7 across all instances, including source file names, system titles, table legends, and manuscript callouts.

References are not permitted in Table EV6; please remove all references from this table. After the other EV tables have been renamed as EV datasets, Table EV6 should be renumbered as Table EV1, and the callouts need to be updated accordingly.

We thank the editors for noticing these discrepancies. All legends within EV datasets and tables have been corrected and references have been removed. All previous EV Tables except Table EV6 have been reclassified as EV Datasets, and Table EV6 has been renumbered to Table EV1. Callouts have been updated accordingly.

Please use our Reagents and Tools Table template (.docx), which you can find in our author guidelines: <https://link.springer.com/journal/44320/submission-guidelines#structuredmethods>.

We have included a separate .docx file with the Reagents and Tools Table and removed the embedded table from the revised manuscript.

Please address the following issues in figure legends:

- Please note that the box plots need to be defined in terms of minima, maxima, centre, bounds of box and whiskers, and percentile in the legends of figures 1D, EV1 B, C
All specified figure legends have been updated with required definitions for boxplots.
- Please note that information related to n is missing in the legends of figures 2C, E, F; EV1 C, EV4 B, C.
The figure legend for EV1 C has been updated with n and other clarifying information. All other figure legends specify n.
- Please note that the error bars are not defined in the legends of figures 2C, E, F; EV4 B, C
Where required, figure legends have been updated to clarify and define the error bars.

Reviewer Comments

Reviewer #2: The authors have clearly and comprehensively addressed feedback. Our concerns have been addressed. This is a strong and important manuscript that deserves publication.

Reviewer #3: The authors have addressed my concerns and I believe this work will be an extremely valuable contribution to the field.

We thank the reviewers for their time, expertise, and thoughtful comments.

8th Jan 2026

Manuscript number: MSB-2025-13376RR

Title: Predicting the protein interaction landscape of a free-living bacterium with pooled-AlphaFold3

Dear Horia,

Thank you again for sending us your revised manuscript. We are now satisfied with the modifications made and I am pleased to inform you that your paper has been accepted for publication.

You may qualify for financial assistance for your publication charges - either via a Springer Nature fully open access agreement or an EMBO initiative. Check your eligibility: <https://link.springer.com/journal/44320/how-to-publish-with-us>

Kind regards,
Jingyi

Jingyi Hou, PhD
Senior Editor
Molecular Systems Biology

>>> Please note that it is Molecular Systems Biology policy for the transcript of the editorial process (containing referee reports and your response letter) to be published as an online supplement to each paper. If you do NOT want this, you will need to inform the Editorial Office via email immediately. More information is available here: <https://link.springer.com/partners/embo-press/editorial-policies#Peer%20review>